# Tight Nonparametric Convergence Rates for Stochastic Gradient Descent under the Noiseless Linear Model

**Raphaël Berthier**
INRIA, Ecole Normale Supérieure
PSL Research University, Paris, France
`raphael.berthier@inria.fr`

**Francis Bach**
INRIA, Ecole Normale Supérieure
PSL Research University, Paris, France
`francis.bach@inria.fr`

**Pierre Gaillard**
INRIA, Ecole Normale Supérieure
PSL Research University, Paris, France
`pierre.gaillard@inria.fr`

## Abstract

In the context of statistical supervised learning, the noiseless linear model assumes that there exists a deterministic linear relation $Y = \langle \theta_*, \Phi(U) \rangle$ between the random output $Y$ and the random feature vector $\Phi(U)$, a potentially non-linear transformation of the inputs $U$. We analyze the convergence of single-pass, fixed step-size stochastic gradient descent on the least-square risk under this model. The convergence of the iterates to the optimum $\theta_*$ and the decay of the generalization error follow polynomial convergence rates with exponents that both depend on the regularities of the optimum $\theta_*$ and of the feature vectors $\Phi(U)$. We interpret our result in the reproducing kernel Hilbert space framework. As a special case, we analyze an online algorithm for estimating a real function on the unit hypercube from the noiseless observation of its value at randomly sampled points; the convergence depends on the Sobolev smoothness of the function and of a chosen kernel. Finally, we apply our analysis beyond the supervised learning setting to obtain convergence rates for the averaging process (a.k.a. gossip algorithm) on a graph depending on its spectral dimension.

## 1 Introduction

Linear regression is widely used in statistical supervised learning, sometimes in the implicit form of kernel regression. A large theory describes the performance (reconstruction and generalization errors) of various algorithms (penalized least-squares, stochastic gradient descent, . . . ) under various data models (e.g., noisy or noiseless linear model) and the corresponding minimax bounds. In nonparametric estimation theory, one seeks bounds independent of the dimension of the underlying feature space [16, 30]: these bounds describe best the observed behavior in many modern linear regressions, where the data are inherently high-dimensional or where a kernel associated to a high-dimensional feature map is used. In this paper, we provide nonparametric bounds for stochastic gradient descent under the noiseless linear model and under small perturbations of this model.

Under the noiseless linear model, we assume that there exists a ground-truth linear relation $Y = \langle \theta_*, X \rangle$ between the feature vector $X$ and the output $Y \in \mathbb{R}$. The feature vector $X$ may be itself a non-linear transformation of the inputs $U$, explicitly computed through a feature map $X = \Phi(U)$ or implicitly defined through a positive-definite kernel $k(U, U')$ [17]. The noiseless linear model

assumes that there exists a linear predictor in feature space with zero generalization error. The difficulty to approximate this optimal prediction rule $\theta_*$ from independent identically distributed (i.i.d.) samples $(X_1, Y_1), \ldots, (X_n, Y_n)$ depends on some measure of the complexity of $\theta_*$.

The noiseless assumption is relevant for some basic vision or sound recognition tasks, where there is no ambiguity of the output $Y$ given the input $U$, but the rule determining the output from the input can be complex. An example from [18, Section 6] is the classification of images of cats versus dogs. For typical images, the output is unambiguous; humans indeed achieve a near-zero error. In sound recognition, one could think of the recovery of the melody from a tune, an unambiguous (but tremendously complex!) task.

Note that in the noiseless model, there is still the randomness of the sampling of $X_1, \ldots, X_n$, sometimes called multiplicative noise because algorithms end up multiplying random matrices [13]. Given those inputs, the outputs $Y_1, \ldots, Y_n$ are deterministic: there is no additive noise, and thus the noiseless linear model we consider in this paper is a simplification of problems with low additive noise.

The large dimension and number of samples in modern datasets motivate the use of first-order online methods [8, 7]. We study the archetype of these methods: single-pass, constant step-size stochastic gradient descent with no regularization, referred to as simply "SGD" in the following.

**Contributions.** Our theoretical results and simulation agree to the following: under the noiseless linear model, the iterates of SGD converge to the optimum $\theta_*$ and the generalization error of SGD vanishes as the number of samples increases. Moreover, the convergence rate of SGD is determined by the minimum of two parameters: the regularity of the optimum $\theta_*$ and the regularity of the feature vectors $X$, where regularities are measured in terms of power norms of the covariance matrix $\Sigma = \mathbb{E}[X \otimes X]$, see Section 2 for precise definitions and statements. Our analysis of the convergence is tight as we prove upper and lower bounds on the performance of SGD that almost match. Thus SGD shows some adaptivity to the complexity of the problem. In Appendix D, we study the robustness of our results when the noiseless linear assumption does not hold, but the generalization error of the optimal linear regression is small. We prove that the asymptotic generalization error of SGD deteriorates by a constant factor, proportional to this optimal generalization error.

Two extensions of our results are studied. First, in Section 3.1, the extension to kernel regression is derived, with, as a special case, the application to the interpolation of a real function $f_*$ on the torus $[0, 1]^d$ from the observation of its value at randomly uniformly sampled points. In the latter case, we show that the rate of convergence depends on the Sobolev smoothness of the function $f_*$ and of the interpolating kernel. Second, beyond supervised learning, our abstract result can be seen as a result on products of i.i.d. linear operators on a Hilbert space. In Section 3.2, we use this result to study a linear stochastic process, the averaging process on a graph, which models a key algorithmic step in decentralized optimization, the gossip algorithm [28, 24]. We prove polynomial convergence rates depending on the spectral dimension of the graph. Finally, in Appendix A, a toy application instantiates our results in the special case of Gaussian features.

**Comparison to the existing literature on linear / kernel regression.** There is an extensive research on the performance of different estimators in nonparametric supervised learning, however almost all of them do not consider the special case of the noiseless linear model [16, 9, 30, 15]. The difference is significant; for instance, rates faster than $O(n^{-1})$ for the least-square risk are impossible with additive noise, while in this paper we prove that SGD can converge with arbitrarily fast polynomial rates. Some of these works analyse the performance of SGD [34, 4, 29, 26, 12, 13, 20, 25, 23]. However, because of the additive noise of the data, convergence requires averaging or decaying step sizes. As a notable exception, [18] studies a variant of kernel regularized least-squares and notices that the rate of convergence improves on noiseless data compared to noisy data. However, their rates are not directly comparable to ours as they assume that the optimal predictor is outside of the kernel space while we focus in Section 3.1 on the attainable case where the optimal predictor is in this space. We make a more precise comparison of this work with our results in Remark 2.

While our work focuses on the test error, a recent trend studies the ability of SGD to reach zero training error in the so-called "interpolation regime", that is in over-parametrized models where a perfect fit on the training data is possible [27, 21, 10]. Even with a fixed step size, SGD is shown to achieve zero-training error. However, these results are significantly different from ours: zero training error does not give any information on the generalization ability of the learned models, and

the "interpolation regime" does not imply the noiseless model. The authors of [31] study a mixed framework that includes both the interpolation regime and the noiseless model, depending on whether SGD is seen as a stochastic algorithm minimizing the generalization error or the training error. An acceleration of SGD is studied, depending on the convexity property of the loss, but not on the nonparametric regularity of the problem.

The field of scattered data approximation [33] studies the estimation of a function from the observation of its values at (possibly random) points, considered in Section 3.1. Again, most of the work focuses on the case where the observation of the values is noisy. We found two exceptions that consider the noiseless case. In [5], a minimax rate of $\Omega((\log n/n)^{p/d})$ is shown for estimating a $p$-smooth function on $[0,1]^d$ in $L^\infty$ norm using $n$ independent uniformly distributed points; the minimax rate is reached with a spline estimate. In [19], a minimax rate of $\Omega(1/n^p)$ in shown for the same problem, but in the special case of $d = 1$ and estimation in $L^1$ norm; the minimax rate is reached with some nearest neighbor polynomial interpolation. Our results are not rigorously comparable with these as we consider the approximation in $L^2$ norm and a definition of smoothness different from theirs. However, roughly speaking, our convergence rate in Section 3.1 of $\Omega(1/n^{1-d/(2p+d)})$ when $p > d/2$ is much slower than theirs. Note that previous estimators could not be computed in an online fashion, and thus have a significantly larger running time. But in general, this suggests that SGD might not achieve the nonparametric minimax rates under the noiseless linear model.

## 2 Linear regression

### 2.1 Setting and main results

We consider the regression problem of learning the linear relationship between a random feature variable $X \in \mathcal{H}$ and a random output variable $Y \in \mathbb{R}$. The feature space $\mathcal{H}$ is assumed to be a Hilbert space with scalar product $\langle .,. \rangle$ and norm $\|.\|$. We assume a noiseless linear model: there exists $\theta_* \in \mathcal{H}$ such that $Y = \langle \theta_*, X \rangle$ almost surely (a.s.). In the online regression setting, we learn $\theta_*$ from i.i.d. observations $(X_1, Y_1), (X_2, Y_2), \ldots$ of $(X, Y)$. SGD proceeds as follows: it starts with the non-informative initialization $\theta_0 = 0$ and at iteration $n$, with current estimate $\theta_{n-1}$, it estimates the risk function on the observation $(X_n, Y_n)$, $\mathcal{R}_n(\theta) = (\langle \theta, X_n \rangle - Y_n)^2/2$ and it performs one step of gradient descent on $\mathcal{R}_n$:

$$
\begin{aligned}
\theta_n &= \theta_{n-1} - \gamma \nabla \mathcal{R}_n(\theta_{n-1}) \\
&= \theta_{n-1} - \gamma \left( \langle \theta_{n-1}, X_n \rangle - Y_n \right) X_n \\
&= \theta_{n-1} - \gamma \langle \theta_{n-1} - \theta_*, X_n \rangle X_n .
\end{aligned}
\tag{1}
$$

The risk $\mathcal{R}_n(\theta)$ is an unbiased estimate of the population risk, also called generalization error,

$$
\mathcal{R}(\theta) = \frac{1}{2} \mathbb{E} \left[ (\langle \theta, X \rangle - Y)^2 \right] = \frac{1}{2} \mathbb{E} \left[ \langle \theta - \theta_*, X \rangle^2 \right] .
$$

We assume the feature variable to be uniformly bounded, namely that there exists a constant $R_0 < \infty$ such that

$$
\|X\|^2 \leqslant R_0 \qquad \text{a.s.}
\tag{2}
$$

We can then define the covariance operator $\Sigma = \mathbb{E}[X \otimes X]$ of $X$, where if $x \in \mathcal{H}$, $x \otimes x$ is the bounded linear operator $\theta \in \mathcal{H} \mapsto \langle \theta, x \rangle x$. Finally, note that,

$$
\mathcal{R}(\theta) = \frac{1}{2} \langle \theta - \theta_*, \Sigma (\theta - \theta_*) \rangle .
$$

We do not assume that the linear operator $\Sigma$ is invertible as this is incompatible in infinite dimension with the boundedness assumption in Eq. (2). Throughout this paper, we use the following convenient notation: if $\alpha$ is a positive real and $\theta$ a vector, $\left\| \Sigma^{-\alpha/2} \theta \right\|^2 = \langle \theta, \Sigma^{-\alpha} \theta \rangle := \inf \left\{ \|\theta'\|^2 \mid \theta' \text{ such that } \theta = \Sigma^{\alpha/2} \theta' \right\}$, with the convention that it is equal to $\infty$ when $\theta \notin \Sigma^{\alpha/2}(\mathcal{H})$. We have two theorems (upper and lower bounds) showing tight convergence rates for SGD.

**Theorem 1** (upper bound). *Assume that there exists a non-negative real number $\underline{\alpha}$ such that*

- *(a) (regularity of the optimum) $\theta_* \in \Sigma^{\underline{\alpha}/2}(\mathcal{H})$, i.e., $\|\Sigma^{-\underline{\alpha}/2}\theta_*\| < \infty$, and*
- *(b) (regularity of the feature vector) $X \in \Sigma^{\underline{\alpha}/2}(\mathcal{H})$ a.s., and there exists a constant $R_{\underline{\alpha}} < \infty$ such that $\|\Sigma^{-\underline{\alpha}/2}X\|^2 \leqslant R_{\underline{\alpha}}$ a.s.*

*Assume further $0 < \gamma \leqslant 1/R_0$. The iterates $\theta_n$ of SGD with step-size $\gamma$ satisfy for all $n \geqslant 1$,*

1. *(reconstruction error)* $$\mathbb{E}\left[\|\theta_n - \theta_*\|^2\right] \leqslant \frac{C}{n^{\underline{\alpha}}},$$

2. *(generalization error)* $$\min_{k=0,\ldots,n} \mathbb{E}\left[\mathcal{R}(\theta_k)\right] \leqslant \frac{C'}{n^{\underline{\alpha}+1}},$$

*where $C = \dfrac{\underline{\alpha}^{\underline{\alpha}}}{\gamma^{\underline{\alpha}}} \left(\|\Sigma^{-\underline{\alpha}/2}\theta_*\|^2 + \dfrac{R_{\underline{\alpha}}}{R_0}\|\theta_*\|^2\right)$ and $C' = 2^{\underline{\alpha}}\dfrac{\underline{\alpha}^{\underline{\alpha}}}{\gamma^{\underline{\alpha}+1}} \left(\|\Sigma^{-\underline{\alpha}/2}\theta_*\|^2 + \dfrac{R_{\underline{\alpha}}}{R_0}\|\theta_*\|^2\right).$*

Assumption (a) is classical in the non-parametric kernel literature [9]: it is often called *complexity of the optimum*, or *source condition*. Assumption (b) is made in [25]. It implies that

$$\mathrm{Tr}(\Sigma^{1-\underline{\alpha}}) = \mathbb{E}[\mathrm{Tr}(XX^T\Sigma^{-\underline{\alpha}})] = \mathbb{E}[X^T\Sigma^{-\underline{\alpha}}X] \leqslant R_{\underline{\alpha}}.$$

This last condition, called *capacity condition* [25], is sometimes stated under the form of a given decay of the eigenvalues of $\Sigma$; it is related to the *effective dimension* of the problem [9].

**Theorem 2** (lower bound). *Assume that there exists a positive real number $\overline{\alpha}$ such that one of the two following conditions holds:*

(a) *(irregularity of the optimum)* $\theta_* \notin \Sigma^{\overline{\alpha}/2}(\mathcal{H})$*, i.e.,* $\|\Sigma^{-\overline{\alpha}/2}\theta_*\| = \infty$*, or*
(b) *(irregularity of the feature vector) with positive probability,* $X \notin \Sigma^{\overline{\alpha}/2}(\mathcal{H})$ *and* $\langle X, \theta_*\rangle \neq 0$*.*

*Assume further $0 < \gamma \leqslant 1/R_0$. The iterates $\theta_n$ of SGD with step-size $\gamma$ satisfy for all $\varepsilon > 0$,*

1. *(reconstruction error)* $\mathbb{E}\left[\|\theta_n - \theta_*\|^2\right]$ *is not asymptotically dominated by $1/n^{\overline{\alpha}+\varepsilon}$,*
2. *(generalization error)* $\mathbb{E}\left[\mathcal{R}(\theta_n)\right]$ *is not asymptotically dominated by $1/n^{\overline{\alpha}+1+\varepsilon}$.*

The take-home message of Theorems 1, 2 is that the convergence rate of SGD is governed by two real numbers: the regularity $\alpha_1$ of the optimum, that is the supremum of all $\underline{\alpha}$ such that $\theta_* \in \Sigma^{\underline{\alpha}/2}(\mathcal{H})$, and the regularity $\alpha_2$ of the features, that is the supremum of all $\underline{\alpha}$ such that $X \in \Sigma^{\underline{\alpha}/2}(\mathcal{H})$ almost surely. The polynomial convergence rate of SGD is roughly of the order of $n^{-\alpha}$ for the reconstruction error and $n^{-\alpha-1}$ for the generalization error with $\alpha = \min(\alpha_1, \alpha_2)$: one of the two regularities is a bottleneck for fast convergence. See Section 3.1 for an application to the optimal choice of a reproducing kernel Hilbert space. The exponent $\alpha_1$ corresponds to the decay of the errors of the gradient descent on the population risk $\mathcal{R}$. However, due to the multiplicative noise, the convergence of SGD is slowed down by the irregularity of the feature vectors if $\alpha_2 < \alpha_1$.

In the theorems, the constraint on the step-size $0 < \gamma \leqslant 1/R_0$ is independent of the time horizon $n$ and of the regularities $\alpha_1, \alpha_2$. Thus fixed step-size SGD shows some adaptivity to the regularity of the problem.

In Section 3, we give extensive numerical evidence that the polynomial rates $n^{-\alpha}$ and $n^{-(\alpha+1)}$ in the bounds are indeed sharp in describing convergence rate of SGD.

We end this section with a few remarks on Theorems 1, 2. They articulate the significance of the results, but are non-essential to the rest of this paper.

**Remark 1.** *Our upper bound and lower bound on the generalization errors do not match exactly. Indeed, we prove an upper bound on the* minimum *risk of the past iterates, where we prove a lower bound on a larger quantity, the risk of the* last *iterate. To the best of our knowledge, it is an open question whether one can prove an upper bound for the last iterate under our assumptions: more precisely, does $\mathbb{E}\left[\mathcal{R}(\theta_n)\right] \leqslant C''/n^{\underline{\alpha}+1}$ hold for some constant $C''$?*

**Remark 2** (related literature). *In the case $\underline{\alpha} = 0$, where no regularity assumption is made on the optimum or the features (apart from being bounded), we upper-bound $\min_{k=1,\ldots,n} \mathbb{E}\left[\mathcal{R}(\theta_k)\right]$ by $O(n^{-1})$. A similar result was shown in [4]: the excess risk for averaged constant-step size SGD is asymptotically dominated by $n^{-1}$ on any least-squares problem–not necessarily a noiseless one. It is remarkable that under the noiseless linear setting, no averaging or decay of the step-size is needed to obtain the same convergence rate.*

*The article [18] also studies the performance of an algorithm, a variant of kernel regularized least-squares, in the noiseless non-parametric setting. However, they do not exploit when the function is more regular than being in the kernel space, i.e., when $\alpha_1 > 0$ with our notation, $\beta > 1/2$ with*

*theirs. In fact, they leave this case as an open problem in their Section 6. Thus, a fair comparison can only be made when $\alpha_1 = 0, \beta = 1/2$. In this case, SGD and the algorithm of [18] both achieve the same rate $O(n^{-1})$.*

**Remark 3.** *The theorems stated above stay true if one weakens the assumptions in the following way, where $\preccurlyeq$ denotes the semi-definite order:*

- *assume $\mathbb{E}\left[\|X\|^2 X \otimes X\right] \preccurlyeq R_0 \Sigma$ instead of $\|X\|^2 \leqslant R_0$ a.s., and*
- *assume $\mathbb{E}\left[\langle X, \Sigma^{-\underline{\alpha}} X \rangle X \otimes X\right] \preccurlyeq R_{\underline{\alpha}} \Sigma$ instead of $\langle X, \Sigma^{-\underline{\alpha}} X \rangle \leqslant R_{\underline{\alpha}}$ a.s.*

*This weaker set of assumptions is useful in the case of non-bounded features, like the Gaussian features of Appendix A. We thus take special care in using only these weaker assumptions in the proofs of Theorems 1, 2, 3 and 4. However we prefer stating results with the stronger assumptions for the sake of clarity.*

**Remark 4** (Application of Theorem 1 in finite dimension). *If $\mathcal{H}$ is finite-dimensional and $\Sigma$ is of full rank, the assumptions of Theorem 1 hold for any $\underline{\alpha} \geqslant 0$. Thus SGD converges faster than any polynomial; in fact one can check that an exponential upper bounds on the reconstruction and generalization errors of the form $C'' \exp(-\lambda_{\min}(\Sigma)t)$ hold, where $\lambda_{\min}(\Sigma)$ is the smallest eigenvalue of $\Sigma$. Although the latter bound is asymptotically better than polynomial rates, for moderate time scales the polynomial rates may describe best the observed behavior; for an illustration of this fact on the averaging process, see Section 3.2 and in particular the discussion following Corollary 1.*

Theorems 1 and 2 are extended in the next section and proved in Appendices B and C respectively. The generalization of Theorem 1 beyond the noiseless linear model is exposed in Appendix D. The reader interested mostly by applications of Theorems 1 and 2 can jump directly to Section 3.

## 2.2 Regularity functions and general results

The main difficulty in the proof of Theorems 1 and 2 is that deriving closed recurrence relations for the expected reconstruction and generalization errors is not straightforward. In this paper, we propose to study the norm of $\theta_n - \theta_*$ associated to different powers of the covariance $\Sigma$. More precisely, define

$$\varphi_n(\beta) = \mathbb{E}\left[\langle \theta_n - \theta_*, \Sigma^{-\beta}(\theta_n - \theta_*) \rangle\right] \in [0, \infty], \qquad \beta \in \mathbb{R}. \tag{3}$$

We call $\varphi_n$ the regularity function at iteration $n$. In particular,

$$\varphi_n(0) = \mathbb{E}[\|\theta_n - \theta_*\|^2] \qquad \text{and} \qquad \varphi_n(-1) = 2\mathbb{E}[\mathcal{R}(\theta_n)].$$

The sequence of regularity functions $\varphi_n$, $n \geqslant 1$ satisfies a closed recurrence inequality (Property 2 in Appendix B) which is central to our proof strategy. Theorems 1 and 2 can be extended to the following estimates on the regularity functions $\varphi_n(\beta)$ on the full interval $\beta \in [-1, \overline{\alpha}]$ (see proofs in Appendices B and C respectively). .

**Theorem 3** (upper bound). *Under the assumptions of Theorem 1, we have for all $n \geqslant 1$,*

*1. for all $\beta \in [0, \underline{\alpha}]$,*
$$\varphi_n(\beta) \leqslant \frac{C}{n^{\underline{\alpha}-\beta}},$$

*2. for all $\beta \in [-1, 0)$,*
$$\min_{k=0,\dots,n} \varphi_k(\beta) \leqslant \frac{C'}{n^{\underline{\alpha}-\beta}},$$

*where $C = \frac{\underline{\alpha}^{\underline{\alpha}-\beta}}{\gamma^{\underline{\alpha}-\beta}}\left(\|\Sigma^{-\underline{\alpha}/2}\theta_*\|^2 + \frac{R_{\underline{\alpha}}}{R_0}\|\theta_*\|^2\right)$, $C' = 2^{\underline{\alpha}-\beta}\frac{\underline{\alpha}^{\underline{\alpha}}}{\gamma^{\underline{\alpha}-\beta}}\left(\|\Sigma^{-\underline{\alpha}/2}\theta_*\|^2 + \frac{R_{\underline{\alpha}}}{R_0}\|\theta_*\|^2\right)$.*

**Theorem 4** (lower bound). *Under the assumptions of Theorem 2, for all $\beta \in [-1, \overline{\alpha}]$, for all $\varepsilon > 0$, $\varphi_n(\beta)$ is not asymptotically dominated by $1/n^{\overline{\alpha}-\beta+\varepsilon}$.*

# 3 Applications

## 3.1 Kernel methods and interpolation in Sobolev spaces

A main case of application of our results is the reproducing kernel Hilbert space (RKHS) setting [17]. In this setting, the space $\mathcal{H}$ is typically large or infinite-dimensional, and we do not have a direct access to the feature variable $X \in \mathcal{H}$. Instead, we have access to some random input variable $U \in \mathcal{U}$

such that $X = \Phi(U)$ for some fixed feature map $\Phi : \mathcal{U} \to \mathcal{H}$. It is then natural to associate a vector $\theta \in \mathcal{H}$ with the function $f_\theta \in L^2(\mathcal{U})$ defined by

$$f_\theta(u) = \langle \theta, \Phi(u) \rangle \, .$$

If the positive-definite kernel $k(u, u') = \langle \Phi(u), \Phi(u') \rangle$ can be computed efficiently, SGD can be "kernelized" [34, 29, 26, 12], i.e., the iteration can be written directly in terms of $f_n := f_{\theta_n}$:

$$f_n = f_{n-1} - \gamma(f_{n-1}(U_n) - Y_n)k(U_n, .)$$
$$= f_{n-1} - \gamma(f_{n-1} - f_*)(U_n)k(U_n, .)$$

where $X_n = \Phi(U_n)$ and $f_*(u) := f_{\theta_*}(u) = \langle \theta_*, \Phi(u) \rangle$. Note that in the kernel literature, the mapping $\theta \mapsto f_\theta$ is used to identify $\mathcal{H}$ with a subspace of $L^2(\mathcal{U})$; indeed, if $\Sigma = \mathbb{E}\left[\Phi(U) \otimes \Phi(U)\right]$ has dense range, the mapping is injective. Using this identification, Theorems 1 and 2 can be applied to obtain bounds in the "attainable" case, meaning that the optimal predictor $f_* \in L^2(\mathcal{U})$ is in the RKHS $\mathcal{H}$. This gives decay rates for the RKHS norm $\|f_n - f_*\| := \|\theta_n - \theta_*\|$ which is inherited from $\mathcal{H}$, but also for the population risk $\mathcal{R}(\theta_n)$ which is reinterpreted as the half squared $L^2$-distance between the associated $f_n$ and the optimal predictor $f_*$. Indeed,

$$\mathcal{R}(\theta_n) = \frac{1}{2}\mathbb{E}\left[\langle \theta_n - \theta_*, \Phi(U) \rangle^2\right] = \frac{1}{2}\mathbb{E}\left[(f_n(U) - f_*(U))^2\right] = \frac{1}{2}\|f_n - f\|^2_{L^2(U)} \, .$$

**Application: interpolation in Sobolev spaces.** To illustrate our results, we consider the case where $\mathcal{U}$ is the torus $[0, 1]^d$, $U$ is uniformly distributed on $\mathcal{U}$ and $k$ is a translation-invariant kernel: $k(u, u') = t(u - u')$ where $t$ is a square-integrable 1-periodic function on $[0, 1]^d$. The kernel $k$ is positive-definite if and only if the Fourier transform of $t$ is positive [32]. This imposes, in particular, that $t$ is maximal at 0. Thus the update rule

$$f_n = f_{n-1} - \gamma \left(f_{n-1}(U_n) - f_*(U_n)\right) t(. - U_n) \tag{4}$$

corrects $f_n$ so that the value $f_n(U_n)$ is closer to the observed value $f_*(U_n)$ than $f_{n-1}(U_n)$. Points near $U_n$ are also updated in the same direction, thus the algorithm should converge rapidly if the function $f_*$ is smooth. Our work derives the polynomial convergence rate as a function of the smoothness of $f_*$ and $t$. The smoothness of functions is measured with the Sobolev spaces $H^s_{\mathrm{per}}$. A function $f$ with Fourier serie $\hat{f}$ belongs to $H^s_{\mathrm{per}}$ if

$$\|f\|^2_{H^s_{\mathrm{per}}} = \sum_{k \in \mathbb{Z}^d} |\hat{f}(k)|^2 \left(1 + |k|^2\right)^s < \infty \, .$$

Assume that the Fourier serie of $t$ satisfies a power-law decay: there exists $c, C > 0$ such that:

$$c\left(1 + |k|^2\right)^{-s/2 - d/4} \leqslant \hat{t}(k) \leqslant C\left(1 + |k|^2\right)^{-s/2 - d/4} \, , \qquad k \in \mathbb{Z}^d \, .$$

This condition does not cover $C^\infty$ kernel, including the Gaussian kernel; it is relevant for less regular kernel, that have a power decay in Fourier. This condition is satisfied, for instance, by the Wendland functions [33, Theorem 10.35], or in dimension $d = 1$ by the kernels corresponding to splines of order $s$, see [32] or [25]. The latter can be computed using the polylogarithm or–for special values of $s$–the Bernoulli polynomials.

We have $t \in H^{s'}_{\mathrm{per}}$ if and only if $s' < s$, thus $s$ measures the Sobolev smoothness of $k$. The operator $\Sigma$ is the convolution with $t$ and thus

$$\|f\|^2 = \langle f, \Sigma^{-1}f \rangle_{L^2} \asymp \sum_{k \in \mathbb{Z}^d} |\hat{f}(k)|^2 \left(1 + |k|^2\right)^{s/2 + d/4} = \|f\|^2_{H^{s/2 + d/4}_{\mathrm{per}}} \, , \tag{5}$$

where $\asymp$ denotes the equality up to positive multiplicative constants. To predict the convergence rate of (4), we check the assumptions of Theorems 1, 2. Computations similar to (5) give

  (a) (regularity of the optimum)

$$\left\langle f_*, \Sigma^{-\alpha}f_* \right\rangle \asymp \left\langle f_*, \Sigma^{-\alpha-1}f_* \right\rangle_{L^2} \asymp \|f_*\|^2_{H^{(s/2 + d/4)(\alpha+1)}_{\mathrm{per}}}$$

  Assume $f_* \in H^r_{\mathrm{per}}$. We have $\left\langle f_*, \Sigma^{-\alpha}f_* \right\rangle < \infty$ if $\alpha \leqslant \frac{2r}{s+d/2} - 1$.

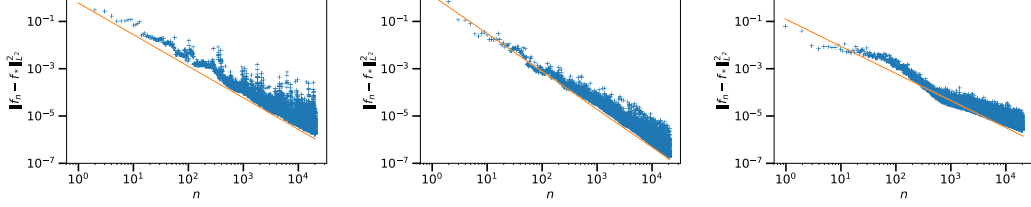

Figure 1: Interpolation of a function of smoothness $r = 2$ using SGD with kernels of smoothness $s = 1$ (left), $s = 2$ (middle) and $s = 3$ (right). Each plot represents one realization of the algorithm (4). The blue crosses represent the square $L^2$ norms $\|f_n - f_*\|_{L^2}^2$ as a function of the number of iterations $n$ and the orange lines represent the predicted polynomials rates $C/n^{\alpha_*+1}$, where $C$ is chosen to match best the empirical observations for each plot.

(b) (regularity of the feature vector)

$$\langle k(u,.), \Sigma^{-\alpha} k(u,.) \rangle = \|k(u,.)\|_{H_{\mathrm{per}}^{(s/2+d/4)(\alpha+1)}}^2 = \|t_s\|_{H_{\mathrm{per}}^{(s/2+d/4)(\alpha+1)}}^2$$

$$= \sum_{k \in \mathbb{Z}^d} (1 + k^2)^{(s/2+d/4)(\alpha-1)}.$$

Thus $\langle k(u,.), \Sigma^{-\alpha} k(u,.) \rangle < \infty$ if and only if $\alpha < 1 - \frac{d}{s+d/2}$.

The regularities of the optimum and of the feature vector are non-negative if the smoothness $s$ of the kernel $t$ satisfies $d/2 < s \leqslant 2r - d/2$, where $r$ is the smoothness of $f_*$. In this case the polynomial rate of decay of the algorithm is given by the exponent

$$\alpha_* = \min\left(\frac{2r}{s+d/2} - 1, 1 - \frac{d}{s+d/2}\right). \tag{6}$$

Note that, given a function $f_*$, this rate is maximal when $s = r$, i.e., the smoothness of the kernel coincides with the smoothness of the function, in which case $\alpha_* = 1 - \frac{d}{r+d/2}$. Theorems 1, 2 give the convergence rates in terms of $L^2$ norm and RKHS norm, which happens to be a Sobolev norm. The more general Theorems 3 and 4 gives convergence rates in terms of a continuity of fractional Sobolev norms, some weaker and some stronger than the RKHS norm.

In Figure 1, we show the decay of the $L^2$ norm in the interpolation of a function $f_*$ on $[0, 1]$ of smoothness 2 using kernels of smaller, matching and larger smoothness. In each case, the rate predicted by (6) is sharp, and the convergence is indeed fastest when the smoothnesses match.

## 3.2 Decay rate of the averaging process

The averaging process is a stochastic process on a graph, mostly studied as a model for asynchronous gossip algorithms on networks. Gossip algorithms are subroutines used to diffuse information throughout networks in distributed algorithms [28], in particular in distributed optimization [24].

Let $G$ be a finite undirected connected graph with vertex set $\mathcal{V}$ of cardinality $N$ and edge set $\mathcal{E}$ of cardinality $M$. The averaging process is a discrete process on functions $x : \mathcal{V} \to \mathbb{R}$ defined as follows. The initial configuration $x_0 = e_{v_\star} : \mathcal{V} \to \mathbb{R}$ is the indicator function of some distinguished vertex $v_\star \in \mathcal{V}$, i.e., $x_0(v_\star) = 1$ and $x_0(v) = 0$ if $v \neq v_\star$. At each iteration, we choose a random edge and replace the values at the ends of the edge by the average of the two current values. In equations, at iterations $n$, given $x_{n-1}$, sample an edge $e_n = \{v_n, w_n\}$ uniformly at random from $\mathcal{E}$ and independently from the past, and define

$$x_n(v_n) = x_n(w_n) = \frac{x_{n-1}(v_n) + x_{n-1}(w_n)}{2}, \qquad x_n(v) = x_{n-1}(v), \quad v \neq v_n, w_n. \tag{7}$$

As the graph is connected, all functions values $x_n(v), v \in \mathcal{V}$ converge to $1/N$ as $n \to \infty$. The study of the averaging process aims at describing how the speed of convergence depends on the graph $G$.

The averaging process can be seen as a prototype interacting particle system, or finite markov information-exchange process according to Aldous's terminology [1]. However, the linear structure

of the updates of the averaging process makes the analysis simpler than in other interacting particle systems; this property is key in applying the results of Section 2.

In this section, we introduce a quantitive version of the notion of spectral dimension of a graph (see [3] and references therein for other definitions). We use this quantity to build polynomial convergence rates for the expected squared $\ell^2$-distance to optimum $\mathbb{E}\left[\sum_{v \in \mathcal{V}}\left(x_n(v)-1/N\right)^2\right]$ and for the expected energy $\mathbb{E}\left[\frac{1}{2}\sum_{\{v,w\}\in\mathcal{E}}(x_n(v)-x_n(w))^2\right]$. The comparison with other known convergence bounds is made. We add numerical experiments showing that our bounds describe the observed behavior in some classical large graphs, for an intermediate number of iterations.

Let $L = \sum_{\{v,w\}\in\mathcal{E}}(e_v - e_w)(e_v - e_w)^\top$ be the Laplacian of the graph. It is a positive semi-definite operator. The spectral measure of $L$ at a vertex $v \in \mathcal{V}$ is the unique measure $\sigma_v$ such that for all continuous real function $f$,

$$\langle e_v, f(L)e_v\rangle = \int \mathrm{d}\sigma_v(\lambda)f(\lambda)\,.$$

If $0 = \lambda_0 < \lambda_1 \leqslant \ldots \leqslant \lambda_{N-1}$ are the eigenvalues of $L$ and $u_0 = \mathbf{1}, u_1, \ldots, u_{N-1}$ are the corresponding normalized eigenvectors, then

$$\sigma_v(\mathrm{d}\lambda) = \sum_{i=0}^{N-1}(u_i(v))^2\delta_{\lambda_i}(\mathrm{d}\lambda)\,.$$

We say that $G$ is of spectral dimension $d \geqslant 0$ with constant $V > 0$ if

$$\forall v \in \mathcal{V}\,, \quad \forall E \in (0,\infty)\,, \quad \sigma_v((0,E]) \leqslant V^{-1}E^{d/2}\,.$$

A typical example motivating this definition is the following.

**Proposition 1.** *Let $\mathbb{T}_\Lambda^d$ denote the d-dimensional torus of side length $\Lambda$, i.e., the graph with vertex set $\mathcal{V} = (\mathbb{Z}/\Lambda\mathbb{Z})^d$ and edge set $\mathcal{E} = \{\{v,w\} \,|\, v,w \in E, \|v-w\|_2 = 1\}$. The torus $\mathbb{T}_\Lambda^d$ is of spectral dimension $d$ with some constant $V(d)$ that depends on the dimension $d$ but not on the side length $\Lambda$.*

This result is proved in Appendix F. Similar results were proved for supercritical percolation bonds in [22] and for the random geometric graphs in [3].

When the graph is large, the probability of sampling a given edge decays to 0. It is natural to define a rescaled time $t = n/M$ so that the expected number of times a given edge is sampled during a unit time interval does not depend on $M$ (and is equal to 1).

**Corollary 1** (of Theorem 1). *Assume that $G$ is of spectral dimension $d$ with constant $V$, and denote $\delta_{\max}$ the maximal degree of the nodes in the graph. Then, for all $t = n/M \geqslant 2$,*

*1.* $$\mathbb{E}\left[\sum_{v\in\mathcal{V}}\left(x_{Mt}(v)-\frac{1}{N}\right)^2\right] \leqslant D(d,V,\delta_{\max})\frac{\log t}{t^{d/2}}\,,$$

*2.* $$\min_{0\leqslant s\leqslant t}\mathbb{E}\left[\frac{1}{2}\sum_{\{v,w\}\in\mathcal{E}}(x_{Ms}(v)-x_{Ms}(w))^2\right] \leqslant D'(d,V,\delta_{\max})\frac{\log t}{t^{d/2+1}}\,,$$

*where $D(d,V,\delta_{\max}) = \dfrac{2}{\log 2}d^{d/2+1}V^{-1}\delta_{\max}$ and $D'(d,V,\delta_{\max}) = \dfrac{2^{d/2+2}}{\log 2}d^{d/2+1}V^{-1}\delta_{\max}$.*

See Appendix E for the proof. Note that as $G$ is a finite graph, $G$ can be of any spectral dimension $d$ for some potentially large constant $V$. However, for many families of graphs of increasing size, such as the toruses $\mathbb{T}_\Lambda^d$, $\Lambda \geqslant 1$, the spectral dimension constant $V$ corresponding to the dimension $d$ and the maximum degree $\delta_{\max}$ remain bounded independently of the size of the graph. In that case, the bounds of Corollary 1 are independent of the size of the graph.

These bounds should be compared to the known exponential convergence bounds of [2] or [28]: they are of the form $O(\exp(-\gamma t))$ where $\gamma$ is the spectral gap of the Laplacian of the graph, the distance between the two minimal eigenvalues of the Laplacian. Although asymptotically faster, these bounds are only relevant on the typical scale $t \gtrsim 1/\gamma$. In many graphs of interests, the spectral gap $\gamma$ vanishes as the size of the graph increases; for instance, when $G = \mathbb{T}_\Lambda^d$, $\gamma$ is of the order of $1/\Lambda^2$. As a consequence, for large graphs and moderate number of iterations, the spectral dimension based

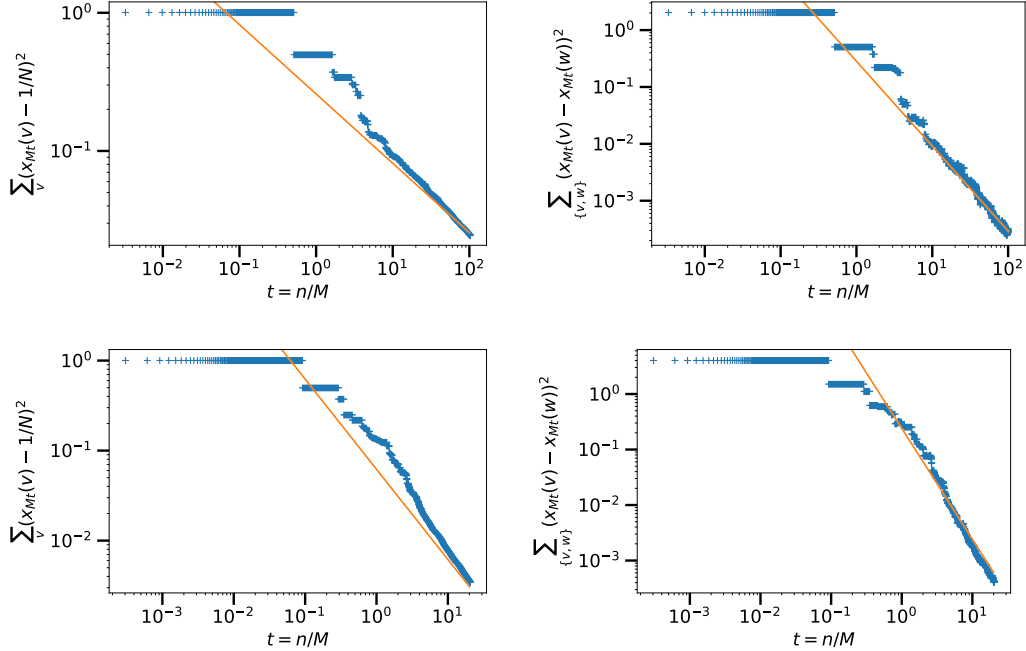

Figure 2: Convergence rates on the circle $\mathbb{T}^1_{300}$ (up) and on the two dimensional torus $\mathbb{T}^2_{40}$ (bottom). The convergence is measured in terms of squared $\ell^2$-distance to $\frac{1}{N}\mathbf{1}$ (left) and sum of the squared differences along the edges (right). In orange are the curves of the form $C/n^{d/2}$ and $C'/n^{d/2+1}$ where $C$ and $C'$ are constants chosen to match best the empirical observations for each plot.

bounds describe the observed behavior where spectral gap based bounds do not apply. Indeed, in Figure 2, simulations on a large circle $\mathbb{T}^1_{300}$ and on a large torus $\mathbb{T}^2_{40}$ display polynomial decay rates, with polynomial exponents coinciding with those of the corresponding bounds of Corollary 1. Note that, if pushed on a longer time scale, the simulations would have shown the exponential convergence due to finite graph effects. This incapacity of spectral gap to describe the transient behavior had already motivated the authors of [6] to use the spectral dimension to describe the behavior and to design accelerations of the gossip algorithm. However, the analyses of this paper control only the expected process $\mathbb{E}[x_n]$: the random sampling of the edges is averaged out.

While the polynomial exponents are sharp, we expect the logarithmic factors to be an artifact of the method of proof.

In the case $d = 0$ and $V = 1$, where no assumption on the structure of the graph is made, the fact that the minimal past energy is $O(n^{-1})$ (neglecting the logarithmic factor) has been noticed by Aldous in [2, Proposition 4]. Aldous leaves as an open problem whether one can prove a bound without taking a minimum; this is a special case of our Remark 1.

## 4 Conclusion and research directions

In this paper, we give a sharp description of the convergence of SGD under the noiseless linear model and made connexions with the interpolation of a real function and the averaging process. The behavior of SGD is surprisingly different in the absence of additive noise: it converges without any averaging or decay of the step-sizes. To some extent, SGD adapts to the regularity of the problem thanks to the implicit regularization ensured by the initialization at zero and the single pass on the data. However, by comparing with some known estimators for the interpolation of functions [5, 19] (see the end of Section 1), we conjecture that the convergence rate of SGD is suboptimal. What are the minimax rates under the noiseless linear model? Can they be reached with some accelerated online algorithm?

## Broader impact

This work does not present any foreseeable societal consequence.

## Acknowledgments

This work was greatly improved by detailed comments from Loucas Pillaud-Vivien on earlier versions of the manuscript. We also thank Alessandro Rudi, Nicolas Flammarion and anonymous reviewers for useful discussions. This work was funded in part by the French government under management of Agence Nationale de la Recherche as part of the "Investissements d'avenir" program, reference ANR-19-P3IA-0001 (PRAIRIE 3IA Institute). We also acknowledge support from the European Research Council (grant SEQUOIA 724063) and from the DGA.

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
