[Supplementary Material]

# A  Linear regression with Gaussian features

In the setting of Section 2.1, we assume $X$ to be centered Gaussian process of covariance $\Sigma$ where $\Sigma$ is a bounded symmetric semidefinite operator. As $X$ is not bounded a.s., we need to use the weaker set of assumptions given in Remark 3. We thus need to compute $R_0$ such that $\mathbb{E}\left[\|X\|^2 X \otimes X\right] \preccurlyeq R_0 \Sigma$ and $\alpha, R_\alpha$ such that $\mathbb{E}\left[\langle X, \Sigma^{-\alpha} X\rangle X \otimes X\right] \preccurlyeq R_\alpha \Sigma$. We show here that these conditions are in fact simple trace conditions on $\Sigma$, sometimes called *capacity conditions* [25].

**Lemma 1.** *If $X \sim \mathcal{N}(0, \Sigma)$ and $A$ is a bounded symmetric operator such that $\mathrm{Tr}(\Sigma A) < \infty$,*

$$\mathbb{E}\left[\langle X, AX\rangle X \otimes X\right] = 2\Sigma A \Sigma + \mathrm{Tr}(\Sigma A)\Sigma \preccurlyeq \left(2\|\Sigma^{1/2} A \Sigma^{1/2}\|_{\mathcal{H} \to \mathcal{H}} + \mathrm{Tr}(\Sigma A)\right)\Sigma.$$

*Proof.* Diagonalize $\Sigma = \sum_{i \geqslant 1} \lambda_i e_i \otimes e_i$. Then there exists independent standard Gaussian random variables $X_i, i \geqslant 0$ such that $X = \sum_i \lambda_i^{1/2} X_i e_i$.

Let $i, j \geqslant 1$.

$$\langle e_i, \mathbb{E}\left[\langle X, AX\rangle X \otimes X\right] e_j\rangle = \mathbb{E}\left[\langle X, AX\rangle \langle e_i, X \otimes X e_j\rangle\right] = \mathbb{E}\left[\langle X, AX\rangle \lambda_i^{1/2} X_i \lambda_j^{1/2} X_j\right]$$
$$= \lambda_i^{1/2}\lambda_j^{1/2} \sum_{k,l} A_{k,l} \lambda_k^{1/2} \lambda_l^{1/2} \mathbb{E}\left[X_i X_j X_k X_l\right].$$

As $X_i, i \geqslant 1$ are centered independent random variables, the quantity $\mathbb{E}\left[X_i X_j X_k X_l\right]$ is 0 in many cases. More precisely,

- if $i \neq j$, the general term of the sum in non-zero only when $k = i$ and $l = j$ or $k = j$ and $l = i$. This gives
$$\langle e_i, \mathbb{E}\left[\langle X, AX\rangle X \otimes X\right] e_j\rangle = 2A_{i,j}\lambda_i\lambda_j.$$

- if $i = j$, the general term of the sum is non-zero only when $k = l$. This gives
$$\langle e_i, \mathbb{E}\left[\langle X, AX\rangle X \otimes X\right] e_i\rangle = \lambda_i \sum_k A_{k,k}\lambda_k \mathbb{E}\left[X_i^2 X_k^2\right] = \lambda_i \sum_{k \neq i} A_{k,k}\lambda_k + 3\lambda_i^2 A_{i,i}$$
$$= \lambda_i \sum_k A_{k,k}\lambda_k + 2\lambda_i^2 A_{i,i}.$$

In both cases,

$$\langle e_i, \mathbb{E}\left[\langle X, AX\rangle X \otimes X\right] e_j\rangle = 2\lambda_i\lambda_j A_{i,j} + \left(\sum_k A_{k,k}\lambda_k\right)\lambda_i \mathbf{1}_{i=j}.$$

Note that

$$\mathrm{Tr}(A\Sigma) = \sum_k \langle e_k, \Sigma A e_k\rangle = \sum_k \lambda_k A_{k,k}.$$

Thus we get

$$\langle e_i, \mathbb{E}\left[\langle X, AX\rangle X \otimes X\right] e_j\rangle = 2\lambda_i\lambda_j A_{i,j} + \mathrm{Tr}(A\Sigma)\lambda_i \mathbf{1}_{i=j}$$
$$= 2\langle e_i, \Sigma A \Sigma e_j\rangle + \mathrm{Tr}(A\Sigma)\langle e_i, \Sigma e_j\rangle$$
$$= \langle e_i, [2\Sigma A \Sigma + \mathrm{Tr}(\Sigma A)\Sigma] e_j\rangle.$$

$\square$

From this lemma with $A = \mathrm{Id}$, we compute $R_0 = 2\|\Sigma\|_{\mathcal{H} \to \mathcal{H}} + \mathrm{Tr}(\Sigma)$, and with $A = \Sigma^{-\alpha}$, we compute $R_\alpha = 2\|\Sigma\|_{\mathcal{H} \to \mathcal{H}}^{1-\alpha} + \mathrm{Tr}(\Sigma^{1-\alpha})$. Thus in the Gaussian case, the condition of (weak) regularity of the features is given by $\mathrm{Tr}(\Sigma^{1-\alpha}) < \infty$.

Figure 3: In blue +, evolution of $\|\theta_n - \theta_*\|^2$ (left) and $\mathcal{R}(\theta_n)$ (right) as functions of $n$, for the problems with parameters $\beta = 1.4, \delta = 1.2$ (up) and $\beta = 3.5, \delta = 1.5$. The orange lines represent the curves $D/n^{\alpha_*}$ (left) and $D'/n^{\alpha_*+1}$ (right).

**Simulations.** We present simulations in finite but large dimension $d = 10^5$, and we check that dimension-independent bounds describe the observed behavior. We artificially generate regression problems with different regularities by varying the decay of the eigenvalues of the covariance $\Sigma$ and varying the decay of the coefficients of $\theta_*$.

Choose an orthonormal basis $e_1, \ldots, e_d$ of $\mathcal{H}$. We define $\Sigma = \sum_{i=1}^d i^{-\beta} e_i \otimes e_i$ for some $\beta \geqslant 1$ and $\theta_* = \sum_{i=1}^d i^{-\delta} e_i$ for some $\delta \geqslant 1/2$. We now check the condition on $\alpha$ such that the assumptions (a) and (b) are satisfied.

(a) $\langle \theta_*, \Sigma^{-\alpha} \theta_* \rangle = \sum_{i=1}^d \langle \theta_*, e_i \rangle^2 i^{\beta\alpha} = \sum_{i=1}^d i^{-2\delta+\alpha\beta}$, which is bounded independently of the dimension $d$ if and only if $\sum_{i=1}^\infty i^{-2\delta+\alpha\beta} < \infty \Leftrightarrow -2\delta + \alpha\beta < -1 \Leftrightarrow \alpha < \frac{2\delta-1}{\beta}$.

(b) $\mathrm{Tr}(\Sigma^{1-\alpha}) = \sum_{i=1}^d i^{-\beta(1-\alpha)}$, which is bounded independently of the dimension $d$ if and only if $\sum_{i=1}^\infty i^{-\beta(1-\alpha)} < \infty \Leftrightarrow -\beta(1-\alpha) < -1 \Leftrightarrow \alpha < 1 - 1/\beta$.

Thus the corollary gives dimension-independent convergence rates for all $\alpha < \alpha_* = \min\left(1 - \frac{1}{\beta}, \frac{2\delta-1}{\beta}\right)$.

In Figure 3, we show the evolution of $\|\theta_n - \theta_*\|^2$ and $\mathcal{R}(\theta_n)$ for two realizations of SGD. We chose the stepsize $\gamma = 1/R_0 = 1/(2\|\Sigma\|_{\mathcal{H}\to\mathcal{H}} + \mathrm{Tr}(\Sigma))$. The two realizations represent two possible different regimes:

- In the two upper plots, $\beta = 1.4, \delta = 1.2$. The irregularity of the feature vectors is the bottleneck for fast convergence. We have $\alpha_* = \min\left(1 - \frac{1}{\beta}, \frac{2\delta-1}{\beta}\right) \approx \min(0.29, 1) = 0.29$.
- In the two lower plots, $\beta = 3.5, \delta = 1.5$. The irregularity of the optimum is the bottleneck for fast convergence. We have $\alpha_* = \min\left(1 - \frac{1}{\beta}, \frac{2\delta-1}{\beta}\right) \approx \min(0.71, 0.57) = 0.57$.

We compare with the curves $D/n^{\alpha_*}$ and $D'/n^{\alpha_*+1}$ with hand-tuned constants $D$ and $D'$ to fit best the data for each plot. In both regimes, our theory is sharp in predicting the exponents in the polynomial rates of convergence of $\|\theta_n - \theta_*\|^2$ and $\mathcal{R}(\theta_n)$.

# B    Proof of Theorems 1 and 3

We recall here the definition of the regularity functions

$$\varphi_n(\beta) = \mathbb{E}\left[\langle \theta_n - \theta_*, \Sigma^{-\beta}(\theta_n - \theta_*)\rangle\right] \in [0, \infty], \qquad \beta \in \mathbb{R}.$$

## B.1    Properties of the regularity functions

We derive here two properties of the sequence of regularity functions $\varphi_n, n \geqslant 1$ that are useful for the proof of Theorem 3. The first one is a simple consequence of the above definition of the regularity function. The second property is the closed recurrence relation of the regularity functions $\varphi_n, n \geqslant 0$ associated to the iterates of SGD.

**Property 1.** *For all $n$, the function $\varphi_n$ is log-convex, i.e., for all $\beta_1, \beta_2 \in \mathbb{R}$, for all $\lambda \in [0, 1]$,*

$$\varphi_n\left((1-\lambda)\beta_1 + \lambda\beta_2\right) \leqslant \varphi_n(\beta_1)^{1-\lambda}\varphi_n(\beta_2)^\lambda.$$

*Proof.* The proof is based on the following lemma, that we state clearly for another use below.

**Lemma 2.** *Let $\theta \in \mathcal{H}$. Then for all $\beta_1, \beta_2 \in \mathbb{R}$, $\lambda \in [0, 1]$,*

$$\left\langle \theta, \Sigma^{-[(1-\lambda)\beta_1 + \lambda\beta_2]}\theta \right\rangle \leqslant \left\langle \theta, \Sigma^{-\beta_1}\theta \right\rangle^{1-\lambda}\left\langle \theta, \Sigma^{-\beta_2}\theta \right\rangle^\lambda.$$

This lemma follows from Hölder's inequality with $p = (1-\lambda)^{-1}$ and $q = \lambda^{-1}$. Indeed, diagonalize $\Sigma = \sum_i \mu_i e_i \otimes e_i$. Then

$$\left\langle \theta, \Sigma^{-[(1-\lambda)\beta_1 + \lambda\beta_2]}\theta \right\rangle = \sum_i \mu_i^{-[(1-\lambda)\beta_1 + \lambda\beta_2]}\langle\theta, e_i\rangle^2$$

$$= \sum_i \left(\mu_i^{-\beta_1}\langle\theta, e_i\rangle^2\right)^{1-\lambda}\left(\mu_i^{-\beta_2}\langle\theta, e_i\rangle^2\right)^\lambda$$

$$\leqslant \left(\sum_i \mu_i^{-\beta_1}\langle\theta, e_i\rangle^2\right)^{1-\lambda}\left(\sum_i \mu_i^{-\beta_2}\langle\theta, e_i\rangle^2\right)^\lambda$$

$$= \left\langle \theta, \Sigma^{-\beta_1}\theta \right\rangle^{1-\lambda}\left\langle \theta, \Sigma^{-\beta_2}\theta \right\rangle^\lambda.$$

We now apply this lemma to prove Property 1.

$$\varphi_n((1-\lambda)\beta_1 + \lambda\beta_2) = \mathbb{E}\left[\left\langle \theta_n - \theta_*, \Sigma^{-[(1-\lambda)\beta_1 + \lambda\beta_2]}(\theta_n - \theta_*)\right\rangle\right]$$

$$\leqslant \mathbb{E}\left[\left\langle \theta_n - \theta_*, \Sigma^{-\beta_1}(\theta_n - \theta_*)\right\rangle^{1-\lambda}\left\langle \theta_n - \theta_*, \Sigma^{-\beta_2}(\theta_n - \theta_*)\right\rangle^\lambda\right].$$

Using again Hölder's inequality, we get

$$\varphi_n((1-\lambda)\beta_1 + \lambda\beta_2) \leqslant \mathbb{E}\left[\left\langle \theta_n - \theta_*, \Sigma^{-\beta_1}(\theta_n - \theta_*)\right\rangle\right]^{1-\lambda}\mathbb{E}\left[\left\langle \theta_n - \theta_*, \Sigma^{-\beta_2}(\theta_n - \theta_*)\right\rangle\right]^\lambda$$

$$= \varphi_n(\beta_1)^{1-\lambda}\varphi_n(\beta_2)^\lambda.$$

$\square$

**Property 2.** *Under the assumptions of Theorem 3, for all $n$, the function $\varphi_n$ is finite on $(-\infty, \underline{\alpha}]$, and if $0 \leqslant \beta \leqslant \underline{\alpha}$,*

$$\varphi_n(\beta) \leqslant \varphi_{n-1}(\beta) - 2\gamma\varphi_{n-1}(\beta - 1) + \gamma^2 R_0^{1-\beta/\underline{\alpha}}R_{\underline{\alpha}}^{\beta/\underline{\alpha}}\varphi_{n-1}(-1).$$

*Proof.* By assumption (a), $\varphi_0(\underline{\alpha}) = \|\Sigma^{-\underline{\alpha}/2}\theta_*\|^2$ is finite, i.e., there exists $\theta \in \mathcal{H}$ such that $\theta_* = \Sigma^{\underline{\alpha}/2}\theta$. Then for any $\beta \leqslant \underline{\alpha}$, $\theta_* = \Sigma^{\beta/2}\left(\Sigma^{(\underline{\alpha}-\beta)/2}\theta\right)$ thus $\varphi_0(\beta) = \|\Sigma^{-\beta/2}\theta_*\|^2$ is finite.

Further, assume that for some $n$, the function $\varphi_{n-1}$ is finite on $(\infty, \underline{\alpha}]$. Then we can rewrite the stochastic gradient iteration (1) as

$$\theta_n - \theta_* = (\mathrm{Id} - \gamma X_n \otimes X_n)(\theta_{n-1} - \theta_*).$$

Substituting this expression in the definition of $\varphi_n$ and expanding the formula, we get

$$\varphi_n(\beta) = \mathbb{E}\left[\langle\theta_n - \theta_*, \Sigma^{-\beta}(\theta_n - \theta_*)\rangle\right]$$
$$= \mathbb{E}\left[\langle(\mathrm{Id} - \gamma X_n \otimes X_n)(\theta_{n-1} - \theta_*), \Sigma^{-\beta}(\mathrm{Id} - \gamma X_n \otimes X_n)(\theta_{n-1} - \theta_*)\rangle\right]$$
$$= \mathbb{E}\left[\langle\theta_{n-1} - \theta_*, \Sigma^{-\beta}(\theta_{n-1} - \theta_*)\rangle\right] \tag{8}$$
$$- 2\gamma\mathbb{E}\left[\langle\theta_{n-1} - \theta_*, \Sigma^{-\beta}X_n \otimes X_n(\theta_{n-1} - \theta_*)\rangle\right] \tag{9}$$
$$+ \gamma^2\mathbb{E}\left[\langle\theta_{n-1} - \theta_*, X_n \otimes X_n\Sigma^{-\beta}X_n \otimes X_n(\theta_{n-1} - \theta_*)\rangle\right]. \tag{10}$$

Note that the first term of this sum is $\varphi_{n-1}(\beta)$. Further, $\theta_{n-1}$ is computed using only $(X_1, Y_1), \ldots, (X_{n-1}, Y_{n-1})$, thus it is independent of $X_n$. It follows that

$$\mathbb{E}\left[\langle\theta_{n-1} - \theta_*, \Sigma^{-\beta}X_n \otimes X_n(\theta_{n-1} - \theta_*)\rangle\right] = \mathbb{E}\left[\langle\theta_{n-1} - \theta_*, \Sigma^{-\beta}\mathbb{E}\left[X_n \otimes X_n\right](\theta_{n-1} - \theta_*)\rangle\right]$$
$$= \mathbb{E}\left[\langle\theta_{n-1} - \theta_*, \Sigma^{-\beta+1}(\theta_{n-1} - \theta_*)\rangle\right]$$
$$= \varphi_{n-1}(\beta - 1). \tag{11}$$

Finally,

$$\mathbb{E}\left[\langle\theta_{n-1} - \theta_*, X_n \otimes X_n\Sigma^{-\beta}X_n \otimes X_n(\theta_{n-1} - \theta_*)\rangle\right] \tag{12}$$
$$= \mathbb{E}\left[\langle\theta_{n-1} - \theta_*, X_n\rangle^2\langle X_n, \Sigma^{-\beta}X_n\rangle\right] \tag{13}$$

We now assume that $0 \leqslant \beta \leqslant \underline{\alpha}$. We apply Lemma 2 with $\beta_1 = 0, \beta_2 = \underline{\alpha}, \lambda = \beta/\underline{\alpha}$:

$$\langle X_n, \Sigma^{-\beta}X_n\rangle \leqslant \|X_n\|^{2(1-\beta/\underline{\alpha})}\langle X_n, \Sigma^{-\underline{\alpha}}X_n\rangle^{\beta/\underline{\alpha}}$$

Let $\mathbb{E}_{X_n}$ denote the expectation with respect to $X_n$ only, while keeping $X_0, \ldots, X_{n-1}$ random. Applying Hölder's inequality, we get

$$\mathbb{E}_{X_n}\left[\langle X_n, \Sigma^{-\beta}X_n\rangle\langle\theta_{n-1} - \theta_*, X_n\rangle^2\right]$$
$$\leqslant \mathbb{E}_{X_n}\left[\|X_n\|^{2(1-\beta/\underline{\alpha})}\langle X_n, \Sigma^{-\underline{\alpha}}X_n\rangle^{\beta/\underline{\alpha}}\langle\theta_{n-1} - \theta_*, X_n\rangle^2\right]$$
$$\leqslant \mathbb{E}_{X_n}\left[\|X_n\|^2\langle\theta_{n-1} - \theta_*, X_n\rangle^2\right]^{1-\beta/\underline{\alpha}}\mathbb{E}\left[\langle X_n, \Sigma^{-\underline{\alpha}}X_n\rangle\langle\theta_{n-1} - \theta_*, X_n\rangle^2\right]^{\beta/\underline{\alpha}}$$
$$= \langle\theta_{n-1} - \theta_*, \mathbb{E}\left[\|X_n\|^2X_n \otimes X_n\right](\theta_{n-1} - \theta_*)\rangle^{1-\beta/\underline{\alpha}}$$
$$\times \langle\theta_{n-1} - \theta_*, \mathbb{E}\left[\langle X_n, \Sigma^{-\underline{\alpha}}X_n\rangle X_n \otimes X_n\right](\theta_{n-1} - \theta_*)\rangle^{\beta/\underline{\alpha}}$$
$$\leqslant R_0^{1-\beta/\underline{\alpha}}R_{\underline{\alpha}}^{\beta/\underline{\alpha}}\langle\theta_{n-1} - \theta_*, \Sigma(\theta_{n-1} - \theta_*)\rangle,$$

where in this last step, we use the assumptions that the features $X$ are bounded and regular, in their weak formulation of Remark 3. Returning to the computation of (12)-(13), we get

$$\mathbb{E}\left[\langle\theta_{n-1} - \theta_*, X_n \otimes X_n\Sigma^{-\beta}X_n \otimes X_n(\theta_{n-1} - \theta_*)\rangle\right]$$
$$= \mathbb{E}\left[\mathbb{E}_{X_n}\left[\langle\theta_{n-1} - \theta_*, X_n\rangle^2\langle X_n, \Sigma^{-\beta}X_n\rangle\right]\right] \tag{14}$$
$$\leqslant R_0^{1-\beta/\underline{\alpha}}R_{\underline{\alpha}}^{\beta/\underline{\alpha}}\mathbb{E}\left[\langle\theta_{n-1} - \theta_*, \Sigma(\theta_{n-1} - \theta_*)\rangle\right]$$
$$= R_0^{1-\beta/\underline{\alpha}}R_{\underline{\alpha}}^{\beta/\underline{\alpha}}\varphi_{n-1}(-1). \tag{15}$$

The result is obtained by putting together Equations (8)-(10), (11) and (15). $\qquad\square$

## B.2 Proof of Theorem 1

A remarkable feature of the proof that follows is that only Properties 1 and 2 of the regularity functions are used to derive the theorem. In particular, we do not use the definition of the regularity functions $\varphi_n$ in this section.

We start with a few preliminary remarks. Using the recurrence Property 2 and that $\gamma R_0 \leqslant 1$,

$$\varphi_k(0) \leqslant \varphi_{k-1}(0) - \gamma\left(2 - \gamma R_0\right)\varphi_{k-1}(-1)$$
$$\leqslant \varphi_{k-1}(0) - \gamma\varphi_{k-1}(-1).$$

Thus the sequence $\varphi_k(0)$, $k \geqslant 0$ decreases, and

$$\gamma\varphi_{k-1}(-1) \leqslant \varphi_{k-1}(0) - \varphi_k(0). \tag{16}$$

By summing this inequality over $k \geqslant 1$, we get

$$\gamma\sum_{k=0}^{\infty}\varphi_k(-1) \leqslant \varphi_0(0). \tag{17}$$

Using again the recurrence Property 2,

$$\varphi_k(\underline{\alpha}) \leqslant \varphi_{k-1}(\underline{\alpha}) - 2\gamma\varphi_{k-1}(\underline{\alpha}-1) + \gamma^2 R_{\underline{\alpha}}\varphi_{k-1}(-1) \tag{18}$$
$$\leqslant \varphi_{k-1}(\underline{\alpha}) + \gamma^2 R_{\underline{\alpha}}\varphi_{k-1}(-1).$$

By summing for $k = 1, \ldots, n$ and using the bound (17),

$$\varphi_n(\underline{\alpha}) \leqslant \varphi_0(\underline{\alpha}) + \gamma^2 R_{\underline{\alpha}}\sum_{k=0}^{n-1}\varphi_k(-1)$$
$$\leqslant \varphi_0(\underline{\alpha}) + \gamma R_{\underline{\alpha}}\varphi_0(0)$$
$$\leqslant \varphi_0(\underline{\alpha}) + \frac{R_{\underline{\alpha}}}{R_0}\varphi_0(0). \tag{19}$$

In words, the sequence $\varphi_n(\underline{\alpha})$, $n \geqslant 0$ is bounded by $D := \varphi_0(\underline{\alpha}) + \frac{R_{\underline{\alpha}}}{R_0}\varphi_0(0)$. As a side note, this proves Theorem 3 for $\beta = \underline{\alpha}$.

We can now give a closed recurrence relation $\varphi_k(0)$, $k \geqslant 0$. Using the log-convexity Property 1,

$$\varphi_{k-1}(0) \leqslant \varphi_{k-1}(-1)^{\underline{\alpha}/(\underline{\alpha}+1)}\varphi_{k-1}(\underline{\alpha})^{1/(\underline{\alpha}+1)} \leqslant \varphi_{k-1}(-1)^{\underline{\alpha}/(\underline{\alpha}+1)}D^{1/(\underline{\alpha}+1)}.$$

Substituting in (16), we obtain

$$\varphi_{k-1}(0) - \varphi_k(0) \geqslant \gamma\varphi_{k-1}(-1)$$
$$\geqslant \gamma D^{-1/\underline{\alpha}}\varphi_{k-1}(0)^{1+1/\underline{\alpha}}.$$

This gives the wanted closed recurrence relation for $\varphi_k(0)$, $k \geqslant 0$. It implies a decay of $\varphi_k(0)$ as follows: consider the real function $f(\varphi) = \frac{1}{\varphi^{1/\underline{\alpha}}}$. It is a convex function on the positive reals, with derivative $f'(\varphi) = -\frac{1}{\underline{\alpha}}\frac{1}{\varphi^{1+1/\underline{\alpha}}}$. Using that a convex function is above its tangents, we obtain

$$f\left(\varphi_k(0)\right) - f\left(\varphi_{k-1}(0)\right) \geqslant f'\left(\varphi_{k-1}(0)\right)\left(\varphi_k(0) - \varphi_{k-1}(0)\right)$$
$$= -\frac{1}{\underline{\alpha}}\frac{1}{\varphi_{k-1}(0)^{1+1/\underline{\alpha}}}\left(\varphi_k(0) - \varphi_{k-1}(0)\right)$$
$$\geqslant \frac{1}{\underline{\alpha}}\gamma D^{-1/\underline{\alpha}}.$$

By summing this inequality for $k = 1, \ldots, n$, we obtain

$$\frac{1}{\varphi_n(0)^{1/\underline{\alpha}}} = f\left(\varphi_n(0)\right) \geqslant f\left(\varphi_0(0)\right) + \frac{1}{\underline{\alpha}}\gamma D^{-1/\underline{\alpha}}n \geqslant \frac{1}{\underline{\alpha}}\gamma D^{-1/\underline{\alpha}}n.$$

This implies conclusion 1 of Theorem 1:

$$\mathbb{E}\left[\|\theta_n - \theta_*\|^2\right] = \varphi_n(0) \leqslant \frac{\underline{\alpha}^{\underline{\alpha}}}{\gamma^{\underline{\alpha}}}D\frac{1}{n^{\underline{\alpha}}}. \tag{20}$$

Further,

$$\min_{0 \leqslant k \leqslant n} \varphi_k(-1) \leqslant \min_{\lceil n/2 \rceil \leqslant k \leqslant n} \varphi_k(-1) \leqslant \frac{2}{n} \sum_{k=\lceil n/2 \rceil}^{n} \varphi_k(-1) \leqslant \frac{2}{n} \frac{1}{\gamma} \sum_{k=\lceil n/2 \rceil}^{n} (\varphi_k(0) - \varphi_{k+1}(0)) \,,$$

where in the last step we used (16). Telescoping the sum, we obtain

$$\min_{0 \leqslant k \leqslant n} \varphi_k(-1) \leqslant \min_{\lceil n/2 \rceil \leqslant k \leqslant n} \varphi_k(-1) \leqslant \frac{2}{n} \frac{1}{\gamma} \varphi_{\lceil n/2 \rceil}(0) \qquad (21)$$

$$\leqslant \frac{2}{n} \frac{1}{\gamma} \frac{\alpha^{\underline{\alpha}}}{\gamma^{\underline{\alpha}}} D \frac{1}{\lceil n/2 \rceil^{\underline{\alpha}}} \leqslant 2^{\underline{\alpha}+1} \frac{\alpha^{\underline{\alpha}}}{\gamma^{\underline{\alpha}+1}} D \frac{1}{n^{\underline{\alpha}+1}} \,.$$

Using that $\varphi_n(-1) = 2\mathbb{E}[\mathcal{R}(\theta_n)]$, this gives conclusion 2 of Theorem 1.

### B.3  Proof of Theorem 3

We continue the proof of Theorem 1 to prove Theorem 3. By the log-convexity Property 1, for all $\beta \in [0, \underline{\alpha}]$,

$$\varphi_n(\beta) \leqslant \varphi_n(0)^{1-\beta/\underline{\alpha}} \varphi_n(\underline{\alpha})^{\beta/\underline{\alpha}} \,.$$

Using Equations (20) and (19), we obtain

$$\varphi_n(\beta) \leqslant \frac{\alpha^{\underline{\alpha}-\beta}}{\gamma^{\underline{\alpha}-\beta}} D \frac{1}{n^{\underline{\alpha}-\beta}} \,.$$

This proves conclusion 1 of the theorem. We now consider the case $\beta \in [-1, 0)$. By the log-convexity Property 1,

$$\min_{0 \leqslant k \leqslant n} \varphi_k(\beta) \leqslant \min_{\lceil n/2 \rceil \leqslant k \leqslant n} \varphi_k(\beta) \leqslant \min_{\lceil n/2 \rceil \leqslant k \leqslant n} \varphi_k(-1)^{-\beta} \varphi_k(0)^{1+\beta}$$

Using that $\varphi_k(0)$, $k \geqslant 0$ is decreasing and the inequality (21), we obtain

$$\min_{\lceil n/2 \rceil \leqslant k \leqslant n} \varphi_k(-1)^{-\beta} \varphi_k(0)^{1+\beta} \leqslant \varphi_{\lceil n/2 \rceil}(0)^{1+\beta} \left( \min_{\lceil n/2 \rceil \leqslant k \leqslant n} \varphi_k(-1) \right)^{-\beta}$$

$$\leqslant \varphi_{\lceil n/2 \rceil}(0)^{1+\beta} \left( \frac{2}{n} \frac{1}{\gamma} \varphi_{\lceil n/2 \rceil}(0) \right)^{-\beta}$$

$$\leqslant \frac{2^{-\beta}}{n^{-\beta}} \frac{1}{\gamma^{-\beta}} \varphi_{\lceil n/2 \rceil}(0) \,.$$

Using finally (20), we obtain conclusion 2 of the theorem

$$\min_{0 \leqslant k \leqslant n} \varphi_k(\beta) \leqslant \frac{2^{-\beta}}{n^{-\beta}} \frac{1}{\gamma^{-\beta}} \frac{\alpha^{\underline{\alpha}}}{\gamma^{\underline{\alpha}}} D \frac{1}{\lceil n/2 \rceil^{\underline{\alpha}}} \leqslant 2^{\underline{\alpha}-\beta} \frac{\alpha^{\underline{\alpha}}}{\gamma^{\underline{\alpha}-\beta}} D \frac{1}{n^{\underline{\alpha}-\beta}} \,.$$

## C  Proof of Theorems 2 and 4

We start in the case (a) where the optimum is irregular: $\theta_* \notin \Sigma^{-\overline{\alpha}/2}(\mathcal{H})$. In that case, we give a lower bound in the convergence rate by studying the expected process $\overline{\theta}_n := \mathbb{E}[\theta_n]$. Indeed, by Jensen's inequality,

$$\varphi_n(\beta) = \mathbb{E}\left[ \langle \theta_n - \theta_*, \Sigma^{-\beta} (\theta_n - \theta_*) \rangle \right] \geqslant \langle \overline{\theta}_n - \theta_*, \Sigma^{-\beta} (\overline{\theta}_n - \theta_*) \rangle \,. \qquad (22)$$

The expectation $\overline{\theta}_n$ can be interpreted as the (non-stochastic) gradient descent on the population risk $\mathcal{R}(\theta)$. Indeed, by taking the expectation in (1), we obtain

$$\overline{\theta}_n - \theta_* = (\text{Id} - \gamma \Sigma)(\overline{\theta}_{n-1} - \theta_*) = -(\text{Id} - \gamma \Sigma)^n \theta_* \,. \qquad (23)$$

Note that as $\gamma \leqslant 1/R_0$, $I - \gamma \Sigma$ is a positive definite matrix. Indeed, by the weak definition of $R_0$ in Remark 3,

$$R_0 \Sigma \succcurlyeq \mathbb{E}\left[ \|X\|^2 X \otimes X \right] = \mathbb{E}\left[ (X \otimes X)(X \otimes X) \right] \succcurlyeq \mathbb{E}[X \otimes X]^2 = \Sigma^2 \,,$$

thus $R_0$ is larger than the operator norm of $\Sigma$. Thus $\gamma\Sigma \preccurlyeq \frac{1}{R_0}\Sigma \preccurlyeq \mathrm{Id}$.

In the following, if $\alpha \in \mathbb{R}$ and $k \in \mathbb{N}$, $\binom{\alpha}{k}$ denotes the generalized binomial coefficient: $\binom{\alpha}{k} = \frac{\alpha(\alpha-1)\cdots(\alpha-k+1)}{k!}$. Fix now $\alpha \geqslant 0$. We have the (formal) power series

$$(1+x)^{-\alpha} = \sum_{k=0}^{\infty} \binom{-\alpha}{k} x^k$$

$$(1-x)^{-\alpha} = \sum_{k=0}^{\infty} \binom{-\alpha}{k} (-1)^k x^k = \sum_{k=0}^{\infty} \binom{\alpha+k-1}{k} x^k$$

$$y^{-\alpha} = \sum_{k=0}^{\infty} \binom{\alpha+k-1}{k} (1-y)^k.$$

This last equality holds in $[0, \infty]$ for $y \in [0, 1]$. In that case, all terms of the serie are positive, thus the meaning of the sum is unambiguous.

Note that $0 \preccurlyeq \gamma\Sigma \preccurlyeq \mathrm{Id}$, thus we have, formally,

$$\gamma^{-\alpha}\Sigma^{-\alpha} = \sum_{k=0}^{\infty} \binom{\alpha+k-1}{k} (\mathrm{Id} - \gamma\Sigma)^k.$$

The rigorous meaning of this equality is that for all $\theta \in \mathcal{H}$,

$$\gamma^{-\alpha}\langle\theta, \Sigma^{-\alpha}\theta\rangle = \sum_{k=0}^{\infty} \binom{\alpha+k-1}{k} \langle\theta, (\mathrm{Id} - \gamma\Sigma)^k\theta\rangle.$$

Both terms of the equality can be infinite: here we are using the convention stated in Section 2.1 that implies that $\langle\theta, \Sigma^{-\alpha}\theta\rangle = \infty \Leftrightarrow \theta \notin \Sigma^{\alpha/2}(\mathcal{H})$. In particular, take $\alpha = \overline{\alpha} - \beta$ and $\theta = \Sigma^{-\beta/2}\theta_*$:

$$\infty = \gamma^{\beta-\overline{\alpha}}\langle\theta_*, \Sigma^{-\overline{\alpha}}\theta_*\rangle = \sum_{k=0}^{\infty} \binom{\overline{\alpha}-\beta+k-1}{k} \langle\theta_*, \Sigma^{-\beta}(\mathrm{Id} - \gamma\Sigma)^k\theta_*\rangle$$

$$= \sum_{n=0}^{\infty} \left[ \binom{\overline{\alpha}-\beta+2n-1}{2n} \langle\theta_*, \Sigma^{-\beta}(\mathrm{Id} - \gamma\Sigma)^{2n}\theta_*\rangle \right.$$

$$\left. + \binom{\overline{\alpha}-\beta+2n}{2n+1} \langle\theta_*, \Sigma^{-\beta}(\mathrm{Id} - \gamma\Sigma)^{2n+1}\theta_*\rangle \right].$$

Using that $\binom{\overline{\alpha}-\beta+2n-1}{2n} \leqslant \binom{\overline{\alpha}-\beta+2n}{2n+1}$ and $\langle\theta_*, \Sigma^{-\beta}(\mathrm{Id} - \gamma\Sigma)^{2n}\theta_*\rangle \geqslant \langle\theta_*, \Sigma^{-\beta}(\mathrm{Id} - \gamma\Sigma)^{2n+1}\theta_*\rangle$ and then (23), (22),

$$\infty \leqslant 2 \sum_{n=0}^{\infty} \binom{\overline{\alpha}-\beta+2n}{2n+1} \langle\theta_*, \Sigma^{-\beta}(\mathrm{Id} - \gamma\Sigma)^{2n}\theta_*\rangle$$

$$= 2 \sum_{n=0}^{\infty} \binom{\overline{\alpha}-\beta+2n}{2n+1} \langle\overline{\theta}_n - \theta_*, \Sigma^{-\beta}(\overline{\theta}_n - \theta_*)\rangle$$

$$\leqslant 2 \sum_{n=0}^{\infty} \binom{\overline{\alpha}-\beta+2n}{2n+1} \varphi_n(\beta).$$

From [14, Equation 5.8.1], we have the formula $\Gamma(z) = \lim_{k\to\infty} \frac{k!k^z}{z(z+1)\cdots(z+k)}$ where $\Gamma$ denotes the Gamma function. Thus as $n \to \infty$

$$\binom{\overline{\alpha}-\beta+2n}{2n+1} = \frac{(\overline{\alpha}-\beta)(\overline{\alpha}-\beta+1)\cdots(\overline{\alpha}-\beta+2n)}{(2n+1)(2n)!} \sim \frac{(2n)^{\overline{\alpha}-\beta}}{(2n+1)\Gamma(\overline{\alpha}-\beta)}.$$

As a consequence, the serie $\sum_n n^{\overline{\alpha}-\beta-1}\varphi_n(\beta)$ diverges. The criteria for the convergence of Riemann series implies that $\varphi_n(\beta)$ can not be asymptotically dominated by $1/n^{\overline{\alpha}-\beta+\varepsilon}$ for $\varepsilon > 0$.

We now turn to the case (b) where the features are irregular: with positive probability $p > 0$, $X \notin \Sigma^{\overline{\alpha}/2}(\mathcal{H})$ and $\langle X, \theta_* \rangle \neq 0$. With probability $p$, the second iterate $\theta_1 = -\gamma \langle X_1, \theta_* \rangle X_1$ is irregular, i.e., $\theta_1 \notin \Sigma^{\overline{\alpha}/2}(\mathcal{H})$. By a simple shift of the iterates, we show that the effect of the irregularity of the initial condition for this iteration started from $\theta_1$ has an effect equivalent to the irregularity of the optimum, thus we can apply the result above to lower bound the convergence rate. More precisely, consider the iterates $\tilde{\theta}_n = \theta_{n+1} - \theta_1$ and $\tilde{\theta}_* = \theta_* - \theta_1$. The iteration (1) can be rewritten as $\tilde{\theta}_n = \tilde{\theta}_{n-1} - \gamma \langle \tilde{\theta}_{n-1} - \tilde{\theta}_*, X_n \rangle X_n$ and $\tilde{\theta}_0 = 0$, thus the new sequence $\tilde{\theta}_n$ satisfies our framework. We can assume that (a) is satisfied, i.e., $\theta_* \in \Sigma^{\overline{\alpha}/2}(\mathcal{H})$. In that case, with probability $p$, $\tilde{\theta}_* = \theta_* - \theta_1 \notin \Sigma^{\overline{\alpha}/2}(\mathcal{H})$. Thus by the case above,

$$\varphi_n(\beta) = \mathbb{E}\left[\langle \theta_n - \theta_*, \Sigma^{-\beta}(\theta_n - \theta_*)\rangle\right]$$
$$= \mathbb{E}\left[\left\langle \tilde{\theta}_{n-1} - \tilde{\theta}_*, \Sigma^{-\beta}\left(\tilde{\theta}_{n-1} - \tilde{\theta}_*\right)\right\rangle\right]$$

is not asymptotically dominated by $1/n^{\overline{\alpha}-\beta+\varepsilon}$, for $\varepsilon > 0$.

## D    Robustness to model mispecification

In this section, we describe how the results of Section 2 are perturbed in the case where a linear relation $Y = \langle \theta_*, X \rangle$ a.s. does not hold. Following the statistical learning framework, we assume a joint law on $(X, Y)$. We further assume that there exists a minimizer $\theta_* \in \mathcal{H}$ of the population risk $\mathcal{R}(\theta)$:

$$\theta_* \in \operatorname*{argmin}_{\theta \in \mathcal{H}}\left\{\mathcal{R}(\theta) = \frac{1}{2}\mathbb{E}\left[(Y - \langle \theta, X \rangle)^2\right]\right\}.$$

This general framework encapsulates two types of perturbations of the noiseless linear model:

- (variance) The output $Y$ can be uncertain given $X$. For instance, under the noisy linear model, $Y = \langle \theta_*, X \rangle + Z$, where $Z$ is centered and independent of $X$. In this case, $\mathcal{R}(\theta_*) = \mathbb{E}[Z^2] = \mathbb{E}[\mathrm{var}\,(Y|X)]$.

- (bias) Even if $Y$ is deterministic given $X$, this dependence can be non-linear: $Y = \psi(X)$ for some non-linear function $\psi$. Then $\mathcal{R}(\theta_*)$ is the squared $L^2$ distance of the best linear approximation to $\psi$: $\mathcal{R}(\theta_*) = \frac{1}{2}\mathbb{E}\left[(\psi(X) - \langle \theta_*, X \rangle)^2\right]$.

In the general framework, the optimal population risk is a combination of both sources

$$\mathcal{R}(\theta_*) = \frac{1}{2}\mathbb{E}\left[\mathrm{var}\,(Y|X)\right] + \frac{1}{2}\mathbb{E}\left[(\mathbb{E}[Y|X] - \langle \theta_*, X \rangle)^2\right].$$

Given i.i.d. realizations $(X_1, Y_1), (X_2, Y_2), \ldots$ of $(X, Y)$, the SGD iterates are defined as

$$\theta_0 = 0, \qquad\qquad \theta_n = \theta_{n-1} - \gamma\left(\langle \theta_{n-1}, X_n \rangle - Y_n\right)X_n. \qquad (24)$$

Apart from the new definition of $\theta_*$, we repeat the same assumptions as in Section 2: let $R_0 < \infty$ be such that $\|X\|^2 \leqslant R_0$ a.s., denote $\Sigma = \mathbb{E}[X \otimes X]$ and $\varphi_n(\beta) = \mathbb{E}\left[\langle \theta_n - \theta_*, \Sigma^{-\beta}(\theta_n - \theta_*)\rangle\right]$.

**Theorem 5.** *Under the assumptions of Theorem 1,*

$$\min_{k=0,\ldots,n} \mathbb{E}\left[\mathcal{R}(\theta_k) - \mathcal{R}(\theta_*)\right] \leqslant 2\frac{C'}{n^{\underline{\alpha}+1}} + 2R_0\gamma\mathcal{R}(\theta_*),$$

*where $C'$ is the same constant as in Theorem 1.*

The take-home message is that if we consider the excess risk $\mathcal{R}(\theta_k) - \mathcal{R}(\theta_*)$, we get the upper bound of the form $2C'n^{-(\underline{\alpha}+1)}$, analog to Theorem 1, but with an additional constant term $2R_0\gamma\mathcal{R}(\theta_*)$. This term can be small if $\mathcal{R}(\theta_*)$ is small, that is if the problem is close to the noiseless linear model, or if the step-size $\gamma$ is small. In the finite horizon setting setting, one can optimize $\gamma$ as a function of the scheduled number of steps $n$ in order to balance both terms in the upper bound. As $C' \propto \gamma^{-(\underline{\alpha}+1)}$, the optimal choice is $\gamma \propto n^{-(\underline{\alpha}+1)/(\underline{\alpha}+2)}$ which gives a rate $\min_{k=0,\ldots,n} \mathbb{E}\left[\mathcal{R}(\theta_k) - \mathcal{R}(\theta_*)\right] = O\left(n^{-(\underline{\alpha}+1)/(\underline{\alpha}+2)}\right)$.

Figure 4: In blue +, evolution of $\|\theta_n - \theta_*\|^2$ (left) and $\mathcal{R}(\theta_n)$ (right) as functions of $n$, for the problems with parameters $d = 10^5, \beta = 1.4, \delta = 1.2$. The orange lines represent the curves $D/n^{\alpha_*}$ (left) and $D'/n^{\alpha_*+1}$ (right).

In the theorem below, we study the SGD iterates $\theta_n$ in terms of the power norms $\varphi_n(\beta)$, $\beta \in [-1, \underline{\alpha} - 1]$, in particular in term of the reconstruction error $\varphi_n(0) = \mathbb{E}[\|\theta_n - \theta_*\|^2]$ if $\underline{\alpha} \geqslant 1$. Note that the population risk $\mathcal{R}(\theta)$ is a quadratic with Hessian $\Sigma$, minimized at $\theta_*$, thus

$$\mathbb{E}\left[\mathcal{R}(\theta_n) - \mathcal{R}(\theta_*)\right] = \frac{1}{2}\mathbb{E}\left[\langle \theta_n - \theta_*, \Sigma(\theta_n - \theta_*)\rangle\right] = \frac{1}{2}\varphi_n(-1).$$

Thus the theorem below extends Theorem 5.

**Theorem 6.** *Under the assumptions of Theorem 1,*

*1. for all $\beta \geqslant 0$, $\beta \leqslant \underline{\alpha} - 1$,*

$$\varphi_n(\beta) \leqslant 2\frac{C(\beta)}{n^{\underline{\alpha}-\beta}} + 4R_0^{1-(\beta+1)/\underline{\alpha}}R_{\underline{\alpha}}^{(\beta+1)/\underline{\alpha}}\gamma\mathcal{R}(\theta_*),$$

*2. for all $\beta \in [-1, 0)$, $\beta \leqslant \underline{\alpha} - 1$,*

$$\min_{k01,\ldots,n} \varphi_k(\beta) \leqslant 2\frac{C'(\beta)}{n^{\underline{\alpha}-\beta}} + 4R_0^{1-(\beta+1)/\underline{\alpha}}R_{\underline{\alpha}}^{(\beta+1)/\underline{\alpha}}\gamma\mathcal{R}(\theta_*),$$

*where $C$, $C'$ are the same constants as in Theorem 3.*

This theorem is proved at the end of this section. We expect the condition $\beta \leqslant \underline{\alpha} - 1$ to be necessary. More precisely, when $\mathcal{R}(\theta_*)$ is positive, we expect the error $\theta_n - \theta_*$ to diverge under the norm $\|\Sigma^{-\beta/2} \cdot \|$ if $\beta > \underline{\alpha} - 1$. In particular, this would imply that the reconstruction error diverges when $\underline{\alpha} < 1$.

In Figure 4, we show how the simulations of Appendix A are perturbed in the presence of additive noise. We consider the noisy linear model $Y = \langle \theta_*, X \rangle + \sigma^2 Z$, where $X \sim \mathcal{N}(0, \Sigma)$ and $Z \sim \mathcal{N}(0, 1)$ are independent. As in the previous simulations, we consider the case $\Sigma = \sum_{i=1}^d i^{-\beta}e_i \otimes e_i$ and $\theta_* = \sum_{i=1}^d i^{-\delta}e_i$ with here $d = 10^5, \beta = 1.4, \delta = 1.2$. In the noiseless case $\sigma^2 = 0$, we have shown that the rate of convergence was given by the polynomial exponent $\alpha_* = \min\left(1 - \frac{1}{\beta}, \frac{2\delta-1}{\beta}\right)$. These predicted rates are represented by the orange lines in the plots. In blue, we show the results of our simulations with some additive noise with variance $\sigma^2 = 2 \times 10^{-4}$. The exponent $\alpha_*$ still describes the behavior of SGD in the initial phase, but in the large $n$ asymptotic the population risk $\mathcal{R}(\theta_n)$ stagnates around the order of $\sigma^2$. Both of these qualitative behaviors are predicted by Theorem 5. Moreover, the reconstruction error $\|\theta_n - \theta_*\|$ diverges for large $n$.

*Proof of Theorems 5 and 6.* Note that in this proof, we use the strong assumptions of regularity of the feature vector $X$. We do not know whether it is possible to prove the same result under the weak assumptions of Remark 3.

Our proof stategy is the following: we decompose the SGD iterates sequence $\theta_n$ as a sum of sequences $\theta_n = \nu_n + \sum_{l=1}^n \eta_n^{(l)}$, where each of the auxiliary sequences is interpreted as the iterates of some

SGD iteration under a noiseless linear model. We thus apply the results of Section 2 to control these auxiliary sequences and obtain the presented bound.

Define $\varepsilon_n = Y_n - \langle \theta_*, X_n \rangle$, the error of the best linear estimator. Then Equation (24) can be rewritten as

$$\theta_0 = 0, \qquad\qquad \theta_n = \theta_{n-1} - \gamma \langle \theta_{n-1} - \theta_*, X_n \rangle X_n + \gamma \varepsilon_n X_n.$$

We see this iteration as an additively perturbed version of the iteration

$$\nu_0 = 0, \qquad\qquad \nu_n = \nu_{n-1} - \gamma \langle \nu_{n-1} - \theta_*, X_n \rangle X_n,$$

studied in Section 2. To understand the effect of the additive noise, define for all $l \geqslant 1$,

$$\eta_l^{(l)} = \gamma \varepsilon_l X_l, \qquad\qquad \eta_n^{(l)} = \eta_{n-1}^{(l)} - \gamma \langle \eta_{n-1}^{(l)}, X_n \rangle X_n, \qquad n > l.$$

Then

$$\theta_n = \nu_n + \sum_{l=1}^{n} \eta_n^{(l)}. \tag{25}$$

Indeed, this last equation is checked by induction: $\theta_0 = 0 = \nu_0$, and if the equation is satisfied for some $n \geqslant 0$,

$$
\begin{aligned}
\theta_{n+1} &= \theta_n - \gamma \langle \theta_n - \theta_*, X_{n+1} \rangle X_{n+1} + \gamma \varepsilon_{n+1} X_{n+1} \\
&= \nu_n + \sum_{l=1}^{n} \eta_n^{(l)} - \gamma \left\langle \nu_n + \sum_{l=1}^{n} \eta_n^{(l)} - \theta_*, X_{n+1} \right\rangle X_{n+1} + \eta_{n+1}^{(n+1)} \\
&= [\nu_n - \gamma \langle \nu_n - \theta_*, X_{n+1} \rangle X_{n+1}] + \sum_{l=1}^{n} \left[ \eta_n^{(l)} - \gamma \langle \eta_n^{(l)}, X_{n+1} \rangle X_{n+1} \right] + \eta_{n+1}^{(n+1)} \\
&= \nu_{n+1} + \sum_{l=1}^{n} \eta_{n+1}^{(l)} + \eta_{n+1}^{(n+1)}.
\end{aligned}
$$

We use the decomposition (25) to study $\varphi_n(\beta)$. Using the triangle inequality,

$$
\begin{aligned}
\varphi_n(\beta) &= \mathbb{E}\left[ \left\| \Sigma^{-\beta/2} \left( \nu_n + \sum_{l=1}^{n} \eta_n^{(l)} \right) \right\|^2 \right] \\
&\leqslant \mathbb{E}\left[ \left( \left\| \Sigma^{-\beta/2} \nu_n \right\| + \left\| \Sigma^{-\beta/2} \sum_{l=1}^{n} \eta_n^{(l)} \right\| \right)^2 \right] \\
&\leqslant 2\mathbb{E}\left[ \left\| \Sigma^{-\beta/2} \nu_n \right\|^2 \right] + 2\mathbb{E}\left[ \left\| \Sigma^{-\beta/2} \sum_{l=1}^{n} \eta_n^{(l)} \right\|^2 \right] \tag{26}
\end{aligned}
$$

The first term is studied in Section 2. We detail the analysis of the second term. Note that

$$
\begin{aligned}
\eta_n^{(l)} &= (I - \gamma X_n \otimes X_n) \eta_{n-1}^{(l)} = \cdots = (I - \gamma X_n \otimes X_n) \cdots (I - \gamma X_{l+1} \otimes X_{l+1}) \eta_l^{(l)} \\
&= (I - \gamma X_n \otimes X_n) \cdots (I - \gamma X_{l+1} \otimes X_{l+1}) \gamma \varepsilon_l X_l. \tag{27}
\end{aligned}
$$

Thus if $l < l'$,

$$
\begin{aligned}
\mathbb{E}\left[ \left\langle \eta_n^{(l)}, \Sigma^{-\beta} \eta_n^{(l')} \right\rangle \right] &= \mathbb{E}\left[ \left\langle \mathbb{E}\left[ \eta_n^{(l)} \Big| X_{l+1}, \ldots, X_n \right], \Sigma^{-\beta} \eta_n^{(l')} \right\rangle \right] \\
&= \mathbb{E}\left[ \left\langle (I - \gamma X_n \otimes X_n) \cdots (I - \gamma X_{l+1} \otimes X_{l+1}) \gamma \mathbb{E}[\varepsilon_l X_l], \Sigma^{-\beta} \eta_n^{(l')} \right\rangle \right]
\end{aligned}
$$

Note that by definition of $\theta_*$, $0 = \nabla \mathcal{R}(\theta_*) = -\mathbb{E}\left[ (Y_l - \langle \theta_*, X_l \rangle) X_l \right] = -\mathbb{E}\left[ \varepsilon_l X_l \right]$ thus we obtain that the cross products $\mathbb{E}\left[ \left\langle \eta_n^{(l)}, \Sigma^{-\beta} \eta_n^{(l')} \right\rangle \right]$ are zero. This gives

$$\mathbb{E}\left[ \left\| \Sigma^{-\beta/2} \sum_{l=1}^{n} \eta_n^{(l)} \right\|^2 \right] = \sum_{l=1}^{n} \mathbb{E}\left[ \left\| \Sigma^{-\beta/2} \eta_n^{(l)} \right\|^2 \right].$$

Note that from Equation (27), $\eta_n^{(l)}$ and $\eta_{n-l+1}^{(1)}$ are equal in law. Thus

$$\mathbb{E}\left[\left\|\Sigma^{-\beta/2}\sum_{l=1}^{n}\eta_n^{(l)}\right\|^2\right] = \sum_{l=1}^{n}\mathbb{E}\left[\left\|\Sigma^{-\beta/2}\eta_{n-l+1}^{(1)}\right\|^2\right] = \sum_{l=1}^{n}\mathbb{E}\left[\left\|\Sigma^{-\beta/2}\eta_l^{(1)}\right\|^2\right]. \quad (28)$$

This last quantity is the sum of the expected squared power norms

$$\varphi_l'(\beta) := \mathbb{E}\left[\left\|\Sigma^{-\beta/2}\eta_l^{(1)}\right\|^2\right]$$

of the SGD iterates $\eta_l^{(1)}, l \geqslant 1$ on a noiseless linear model, with initialization $\eta_1^{(1)} = \gamma\varepsilon_1 X_1$. When $\beta = -1$, this control is given by (17): with our notation here, this gives

$$\sum_{l=1}^{n}\varphi_l'(-1) \leqslant \sum_{l=1}^{\infty}\varphi_l'(-1) \leqslant \frac{1}{\gamma}\varphi_1'(0). \quad (29)$$

When $\beta = \underline{\alpha} - 1$, a similar control can be obtained from (18) which gives:

$$2\gamma\varphi_{l-1}'(\underline{\alpha}-1) \leqslant \varphi_{l-1}'(\underline{\alpha}) - \varphi_l'(\underline{\alpha}) + \gamma^2 R_{\underline{\alpha}}\varphi_{l-1}'(-1).$$

By summing these inequalities for $l = 2, 3, \ldots$, we obtain,

$$2\gamma\sum_{l=1}^{\infty}\varphi_l'(\underline{\alpha}-1) \leqslant \varphi_1'(\underline{\alpha}) + \gamma^2 R_{\underline{\alpha}}\sum_{l=1}^{\infty}\varphi_l'(-1)$$

$$\leqslant \varphi_1'(\underline{\alpha}) + \frac{R_{\underline{\alpha}}}{R_0}\varphi_1'(0) \quad (30)$$

Note that using the strong assumption of regularity of the feature vectors,

$$\varphi_1'(0) = \mathbb{E}\left[\|\gamma\varepsilon_1 X_1\|^2\right] \leqslant \gamma^2 R_0\mathbb{E}\left[\varepsilon_1^2\right] = 2\gamma^2 R_0\mathcal{R}(\theta_*),$$

$$\varphi_1'(\underline{\alpha}) = \mathbb{E}\left[\left\|\Sigma^{-\underline{\alpha}/2}\gamma\varepsilon_1^2 X\right\|^2\right] \leqslant \gamma^2 R_{\underline{\alpha}}\mathbb{E}\left[\varepsilon_1^2\right] = 2\gamma^2 R_{\underline{\alpha}}\mathcal{R}(\theta_*).$$

We use these expressions to simply further (29) and (30):

$$\sum_{l=1}^{n}\varphi_l'(-1) \leqslant 2\gamma R_0\mathcal{R}(\theta_*),$$

$$\sum_{l=1}^{\infty}\varphi_l'(\underline{\alpha}-1) \leqslant 2\gamma R_{\underline{\alpha}}\mathcal{R}(\theta_*).$$

If $\beta \in [-1, \underline{\alpha}-1]$, we use the log-convexity Property 1 and Hölder's inequality: decompose $\beta = (1-\lambda)(-1) + \lambda(\underline{\alpha}-1)$ with $\lambda = (\beta+1)/\underline{\alpha}$,

$$\sum_{l=1}^{\infty}\varphi_l'(\beta) \leqslant \sum_{l=1}^{\infty}\varphi_l'(-1)^{1-\lambda}\varphi_l'(\underline{\alpha}-1)^{\lambda}$$

$$\leqslant \left(\sum_{l=1}^{n}\varphi_l'(-1)\right)^{1-\lambda}\left(\sum_{l=1}^{\infty}\varphi_l'(\underline{\alpha}-1)\right)^{\lambda}$$

$$\leqslant (2\gamma R_0\mathcal{R}(\theta_*))^{1-\lambda}\left(2\gamma R_{\underline{\alpha}}\mathcal{R}(\theta_*)\right)^{\lambda}$$

$$= 2\gamma R_0^{1-\lambda}R_{\underline{\alpha}}^{\lambda}\mathcal{R}(\theta_*). \quad (31)$$

Putting back together Equations (26), (28) and (31), we obtain

$$\varphi_n(\beta) \leqslant 2\mathbb{E}\left[\left\|\Sigma^{-\beta/2}\nu_n\right\|^2\right] + 4\gamma R_0^{1-\lambda}R_{\underline{\alpha}}^{\lambda}\mathcal{R}(\theta_*)$$

The theorem follows the application of Theorem 3 to the sequence $\nu_n$ in order to control the first term. $\qquad\square$

# E Proof of Corollary 1

We apply Theorem 1 in the following way. Denote $\theta_n = x_n - x_0$, $\theta_* = x_* - x_0$, where $x_* = \frac{1}{N}\mathbf{1}$ is the function identically equal to $\frac{1}{N}$. These vectors belong to the Hilbert space $\mathcal{H} = \ell^2(\mathcal{V})$. Denote $\langle .,. \rangle$ and $\|.\|$ the $\ell^2(\mathcal{V})$ scalar product and norm. Denote also $X_n = e_{v_n} - e_{w_n} \in \mathcal{H}$ and $\gamma = 1/2$. Note that $\Sigma = \mathbb{E}[X_n X_n^\top] = \frac{1}{M}L$. The graph is connected thus $\lambda_0 = 0$ is the unique zero eigenvalue of $L$ [11, Lemma 1.7]. The corresponding eigenspace is the space of constant functions. The vectors $\theta_n, X_n, \theta_*$ are orthogonal to the null space of $\Sigma$, thus the quantities of the form $\langle \theta_n, \Sigma^{-\alpha}\theta_n \rangle$, $\langle X_n, \Sigma^{-\alpha}X_n \rangle, \langle \theta_*, \Sigma^{-\alpha}\theta_* \rangle$ are finite.

We have $\theta_0 = 0$ and the averaging update step (7) can be written as

$$\theta_n = \theta_{n-1} - \gamma \left\langle \theta_{n-1} - \theta_*, X_n \right\rangle X_n .$$

The last form makes explicit the parallel with Equation (1). To apply Theorem 1, we check that its assumptions are satisfied. First, $\|X_n\|^2 = 2$ a.s. thus can take $R_0 = 2$ and then $\gamma = 1/R_0$. Second, we seek $\alpha > 0$ such that $\|\Sigma^{-\alpha/2}\theta_*\| < \infty$ and $R_\alpha = \sup_{\{v,w\} \in \mathcal{E}} \langle e_v - e_w, \Sigma^{-\alpha}(e_v - e_w) \rangle < \infty$. In the following, we bound these constants for all $\alpha < d/2$, thus giving decay rates for the expected squared distance to optimum of the form $n^{-\alpha}$ for all $\alpha < d/2$. However, our bounds of the constants $\|\Sigma^{-\alpha/2}\theta_*\|$ and $R_\alpha$ diverge as $\alpha \to d/2$. Nevertheless, by estimating how fast the bounds diverge as $\alpha \to d/2$, we obtain a decay rate of $n^{-d/2}$ by paying an additional logarithmic factor.

Fix $0 < \alpha < d/2$. We check assumptions (a) and (b).

(a)

$$\|\Sigma^{-\alpha/2}\theta_*\|^2 = M^\alpha \left\langle x_* - x_0, L^{-\alpha}(x_* - x_0) \right\rangle = M^\alpha \sum_{i=1}^{N-1} \lambda_i^{-\alpha} \left\langle x_* - x_0, u_i \right\rangle^2 .$$

First, as $x_*$ is a constant vector, $\langle x_*, u_i \rangle$ is zero for all $i \geqslant 1$. Second, $x_0 = e_{v_*}$. Thus

$$
\begin{aligned}
\|\Sigma^{-\alpha/2}\theta_*\|^2 &= M^\alpha \sum_{i=1}^{N-1} \lambda_i^{-\alpha} u_i(v_*)^2 \\
&= M^\alpha \int_{(0,\infty)} \mathrm{d}\sigma_{v_*}(\lambda)\, \lambda^{-\alpha} \\
&= M^\alpha \int_{(0,\infty)} \mathrm{d}\sigma_{v_*}(\lambda) \int_0^\infty \mathrm{d}s\, \mathbf{1}_{\{s \leqslant \lambda^{-\alpha}\}} \\
&= M^\alpha \int_0^\infty \mathrm{d}s \int_{(0,\infty)} \mathrm{d}\sigma_{v_*}(\lambda)\, \mathbf{1}_{\{\lambda \leqslant s^{-1/\alpha}\}} \\
&= M^\alpha \int_0^\infty \mathrm{d}s\, \sigma_{v_*}((0, s^{-1/\alpha}]) .
\end{aligned}
$$

The graph $G$ is of spectral dimension $d$ with constant $V$, thus $\sigma_{v_*}((0, s^{-1/\alpha}]) \leqslant V^{-1}s^{-\frac{d}{2\alpha}}$. However, if $s < \delta_{\max}^{-\alpha}$, it is better to use a more naive bound. As all eigenvalues of $L$ are smaller or equal than $\delta_{\max}$, $\sigma_{v_*}((0, s^{-1/\alpha}]) \leqslant \sigma_{v_*}((0, \delta_{\max}]) \leqslant V^{-1}\delta_{\max}^{d/2}$. Then

$$\|\Sigma^{-\alpha/2}\theta_*\|^2 \leqslant M^\alpha \left[ \int_0^{\delta_{\max}^{-\alpha}} \mathrm{d}s\, V^{-1}\delta_{\max}^{d/2} + \int_{\delta_{\max}^{-\alpha}}^\infty \mathrm{d}s\, V^{-1}s^{-\frac{d}{2\alpha}} \right]$$

$$= M^\alpha V^{-1}\delta_{\max}^{d/2-\alpha} \frac{d}{d - 2\alpha} .$$

(b) Let $\{v,w\} \in E$. As $\|\Sigma^{-\alpha/2}.\|$ is a norm, by the triangle inequality,

$$
\begin{aligned}
\|\Sigma^{-\alpha/2}(e_v - e_w)\|^2 &= \|\Sigma^{-\alpha/2}[(x_* - e_w) - (x_* - e_v)]\|^2 \\
&\leqslant \left( \|\Sigma^{-\alpha/2}(x_* - e_w)\| + \|\Sigma^{-\alpha/2}(x_* - e_v)\| \right)^2 \\
&\leqslant 2 \left( \|\Sigma^{-\alpha/2}(x_* - e_w)\|^2 + \|\Sigma^{-\alpha/2}(x_* - e_v)\|^2 \right) .
\end{aligned}
$$

We bound the two quantities as above. We obtain

$$R_\alpha = \sup_{v,w \in E} \|\Sigma^{-\alpha/2}(e_v - e_w)\|^2 \leqslant 2M^\alpha V^{-1} \delta_{\max}^{d/2-\alpha} \frac{d}{d-2\alpha} .$$

Theorem 1 gives

$$\mathbb{E}\left[\|x_n - x_*\|^2\right] = \mathbb{E}\left[\|\theta_n - \theta_*\|^2\right] \leqslant \frac{\alpha^\alpha}{\gamma^\alpha}\left(\|\Sigma^{-\alpha/2}\theta_*\|^2 + \frac{R_\alpha}{R_0}\|\theta_*\|^2\right)\frac{1}{n^\alpha}$$

$$\leqslant \frac{(d/2)^\alpha}{(1/2)^\alpha}\left(M^\alpha V^{-1}\delta_{\max}^{d/2-\alpha}\frac{d}{d-2\alpha} + M^\alpha V^{-1}\delta_{\max}^{d/2-\alpha}\frac{d}{d-2\alpha}\|\theta_*\|^2\right)\frac{1}{n^\alpha}$$

Note that $\|\theta_*\|_2^2 \leqslant 1$ and recall the scaling $t = n/M$:

$$\mathbb{E}\left[\|x_n - x_*\|^2\right] \leqslant d^{d/2+1}V^{-1}\delta_{\max}^{d/2-\alpha}\frac{1}{d/2-\alpha}\frac{1}{t^\alpha} .$$

This bound is valid for all $\alpha < \frac{d}{2}$. Choose $\alpha = \frac{d}{2} - \frac{\log 2}{\log t}$.

$$\mathbb{E}\left[\|x_n - x_*\|^2\right] \leqslant d^{d/2+1}V^{-1}\delta_{\max}^{\log 2/\log t}\frac{\log t}{\log 2}\frac{2}{t^{d/2}}$$

As we assume $t \geqslant 2$, $\delta_{\max}^{\log 2/\log t} \leqslant \delta_{\max}$. Thus we obtain conclusion 1.

The proof of 2 is similar. Theorem 1 gives

$$\min_{0\leqslant k\leqslant n} \mathbb{E}\left[\frac{1}{2}\sum_{\{v,w\}\in\mathcal{E}}(x_k(v) - x_k(w))^2\right] = \min_{0\leqslant k\leqslant n}\mathbb{E}\left[\frac{1}{2}\langle x_k - x_*, L(x_k - x_*)\rangle\right]$$

$$= M\min_{0\leqslant k\leqslant n}\mathbb{E}\left[\frac{1}{2}\langle\theta_k - \theta_*, \Sigma(\theta_k - \theta_*)\rangle\right]$$

$$\leqslant 2^\alpha \frac{\alpha^\alpha}{\gamma^{\alpha+1}}\left(\|\Sigma^{-\alpha/2}\theta_*\|^2 + \frac{R_\alpha}{R_0}\|\theta_*\|^2\right)\frac{1}{n^\alpha}$$

$$\leqslant 2^{\alpha+1}d^\alpha V^{-1}\delta_{\max}^{d/2-\alpha}\frac{d}{d/2-\alpha}\frac{1}{t^{\alpha+1}} .$$

Taking again $\alpha = \frac{d}{2} - \frac{1}{2\log t}$ and $t \geqslant 2$,

$$\min_{0\leqslant k\leqslant n}\mathbb{E}\left[\frac{1}{2}\sum_{\{v,w\}\in\mathcal{E}}(x_k(v) - x_k(w))^2\right] \leqslant 2^{d/2+1}d^{d/2}V^{-1}\delta_{\max}\frac{d\log t}{\log 2}\frac{2}{t^{d/2+1}}$$

This gives conclusion 2 of the corollary.

## F  Proof of Proposition 1

The graph $\mathbb{T}_\Lambda^d$ is invariant by translation, thus the spectral measure $\sigma_v$ is the same for all vertices $v \in \mathcal{V}$. Thus

$$|\mathcal{V}|\sigma_v(\mathrm{d}\lambda) = \sum_{w\in\mathcal{V}}\sigma_w(\mathrm{d}\lambda) = \sum_{w\in\mathcal{V}}\sum_{i=0}^{N-1}u_i(w)^2\delta_{\lambda_i} = \sum_{i=0}^{N-1}\left(\sum_{w\in\mathcal{V}}u_i(w)^2\right)\delta_{\lambda_i} = \sum_{i=0}^{N-1}\delta_{\lambda_i} .$$

Thus

$$\sigma_v((0, E]) = \frac{1}{\Lambda^d}\left|\{0 < i \leqslant N-1|\lambda_i \leqslant E\}\right| .$$

We need to bound the number of eigenvalues of the Laplacian of $\mathbb{T}_\Lambda^d$ below some fixed value $E$. The eigenvalues of the Laplacian of the circle $\mathbb{T}_\Lambda^1$ are $1 - \cos\left(\frac{2\pi i}{\Lambda}\right)$, $i \in \mathbb{Z}$, $-\Lambda/2 < i \leqslant \Lambda/2$ [11,

Example 1.5]. As $\mathbb{T}_\Lambda^d$ is the Cartesian product $\mathbb{T}_\Lambda^1 \times \cdots \times \mathbb{T}_\Lambda^1$ (with $d$ terms), the eigenvalues of the Laplacian of the torus $\mathbb{T}_\Lambda^d$ are the

$$1 - \cos\left(\frac{2\pi i_1}{\Lambda}\right) + \cdots + 1 - \cos\left(\frac{2\pi i_d}{\Lambda}\right), \qquad i_1, \ldots i_d \in \mathbb{Z}, \quad -\frac{\Lambda}{2} < i_1, \ldots, i_d \leqslant \frac{\Lambda}{2}.$$

For $y \in [-\pi, \pi]$, $1 - \cos(y) \geqslant \frac{2}{\pi^2} y^2$. Thus

$$1 - \cos\left(\frac{2\pi i_1}{\Lambda}\right) + \cdots + 1 - \cos\left(\frac{2\pi i_d}{\Lambda}\right) \leqslant E \Rightarrow \frac{2}{\pi^2}\left[\left(\frac{2\pi i_1}{\Lambda}\right)^2 + \cdots + \left(\frac{2\pi i_d}{\Lambda}\right)^2\right] \leqslant E$$

$$\Leftrightarrow i_1^2 + \cdots + i_d^2 \leqslant \frac{E\Lambda^2}{8}.$$

We need to count the number of integer points in the Euclidean ball centered at $0$ and of radius $\sqrt{E/8}\Lambda$ in $\mathbb{R}^d$. This problem is famously known as Gauss circle problem. For our purposes, a crude estimate suffices: there exists a constant $C(d)$, depending only on the dimension $d$, such that for all radius $R$, the number of integer points in the ball of radius $R$ is smaller than $1 + C(d)R^d$. This leads to the final estimate

$$\sigma_v((0, E]) = \frac{1}{\Lambda^d}\left|\left\{(i_1, \ldots, i_d) \in \left(\mathbb{Z} \cap \left(-\frac{\Lambda}{2}, \frac{\Lambda}{2}\right]\right)^d \setminus \{0\} \text{ such that}\right.\right.$$

$$\left.\left. 1 - \cos\left(\frac{2\pi i_1}{\Lambda}\right) + \cdots + 1 - \cos\left(\frac{2\pi i_d}{\Lambda}\right) \leqslant E\right\}\right|$$

$$\leqslant \frac{1}{\Lambda^d}\left|\left\{(i_1, \ldots, i_d) \in \mathbb{Z}^d \setminus \{0\} \,\middle|\, i_1^2 + \cdots + i_d^2 \leqslant \frac{E\Lambda^2}{8}\right\}\right|$$

$$\leqslant \frac{1}{\Lambda^d}C(d)\left(\frac{E\Lambda^2}{8}\right)^{d/2} = \frac{C(d)}{8^{d/2}}E^{d/2}.$$

This proves the proposition with $V(d) = 8^{d/2}/C(d)$.