[Reviews · NeurIPS 2020]

Review 1

Summary and Contributions: The paper discusses a case of a linear regression model in a Hilbert space, when the regressors are random, but there is no additive noise present. It applies the stochastic gradient iterations for optimizing the objective. While the existence of the additive noise implies well known convergence rates, the paper shows that under the noiseless model the rate of convergence can be significantly higher. The simulation experiments justify the theoretical results.

Strengths: In a field of well-known algorithms and established results, the paper introduces a new view on the same problem. It is important to understand the problem of linear regression better. The focus on infinite-dimensional spaces makes the paper results applicable to kernel regression problems. The upper and lower convergence bounds are established for the expected value of the covariance-weighted risk as well as for the estimation error. The paper shows a simulated example of function interpolation in a high-dimensional space to illustrate the theoretical bounds with the observed convergence rates, which appear to be very close.

Weaknesses: While the paper is written very clearly, there are several questions I’d like to raise. Firstly, in discussing the applicability of the results the paper mentions ‘some basic vision or sound recognition tasks’ (line 33) - I’d like to ask about some examples of such tasks. Looking at the statement of the Theorem 1, seems that it should be applicable in finite-dimensional spaces with invertible covariance matrices. If it is so, then I do not understand the results. In particular, for X distributed with a finite support and has identity covariance matrix, the conditions (a) and (b) hold for arbitrarily large positive \alpha, however the theorem statement implies that the estimates will go to zero at an arbitrarily large polynomial rate, which is not true. The paper does not give any theoretical argument to the 'tightness' of the proposed bounds, so I'd suggest to change the title. In a line 45, one should correct the formulas as Tr() = Tr(X X^T \Sigma^{-\alpha}) = E X^T \Sigma^{-\alpha}X. The product notation in the line 47 is not introduced.

Correctness: There is one question to the statement of the theorem I’d like to be answered.

Clarity: The text is clear and easy to understand.

Relation to Prior Work: The paper does not describe the related work in full details, which is understandable, because linear regression is a well-studied topic. However, I would suggest to add the results with additive noise assumption for infinite-dimensional spaces, to put the proposed model into a perspective.

Reproducibility: Yes

Additional Feedback:


Review 2

Summary and Contributions: This work studies convergence rates for SGD in RKHSs under the noiseless setting and derives the faster convergence rates than $O(1/n)$. ----- After reading the response. The authors have addressed my concerns. I would like to keep my score.

Strengths: Traditionally, the convergence rates of SGD in terms of generalization are studied in a noisy setting except for a few studies. In their analyses, it is known that slower convergence rates than $O(1/n)$ are optimal in general. That is, this paper shows the acceleration of the convergence thanks to the low noise on the regression problems.

Weaknesses: Although the contribution in this work is significant as commented above, I think the claim about the Sobolev smoothness is overstatement because it depends on a somewhat strong condition (line 215-216) which strongly restricts the class of kernels. If there are any misunderstandings in my understanding, I'd appreciate it if you could mention them.

Correctness: The claims seem correct, but I skipped detailed proofs.

Clarity: The paper is well written.

Relation to Prior Work: The difference from the previous studies is well discussed. Additionally, it would be nice if the authors could explain the differences from [18] which also shows the acceleration in the low noise settings.

Reproducibility: Yes

Additional Feedback: As is well known, the averaging technique is known to stabilize and accelerate the convergence of SGD in RKHSs. How does the averaging technique work in the noiseless setting?


Review 3

Summary and Contributions: The paper analyzes the convergence of SGD for the noiseless linear regression model with non-linear transformation on the data. The authors claim that under the noiseless linear regression model, the generalization error of SGD vanishes as the number of sample increases. Then, the authors provide upper bound and lower bounds for the performance of SGD with a small gap. Post Rebuttal: The authors have acknowledged my concerns and I would like to keep my current score.

Strengths: The strength of this work lies in the (almost) matching upper and lower bounds it provides for noiseless linear regression. These results do not require any strong assumptions and can be achieved by assuming regularity of the estimator as well as the data. The experiments in this paper also back up their theoretical claims about the possibility to achieve polynomial convergence rates.

Weaknesses: It seems that [18] does not require the optimal predictor to lie entirely outside the kernel space. In that case, how do the bounds hold up against 18, when the predictor is in the kernel space?

Correctness: The claims seem correct

Clarity: Yes

Relation to Prior Work: The authors do a good job of differentiating their work from previous contributions

Reproducibility: No

Additional Feedback: Line 35: Citation for when it is called multiplicative noise would be great Line 306: Typo connections


Review 4

Summary and Contributions: This paper analyses the convergence rate of stochastic gradient descent in the deceptively simple case of zero noise and square loss in which there exists a deterministic linear relationship between random feature vectors (not necessarily linear in the inputs) and random outputs. It is shown that the convergence rates is given by the minimum between a parameter determining the optimum and the power norm of the covariance matrix (taken as a measure of the regularity of the feature vectors). Theoretical and empirical evidence of the robustness of the findings with respect to the case in which the generalisation error is not negligible is also provided.

Strengths: The results presented in the paper are theoretically sound. Simulations do corroborate the findings. It sheds light on how the convergence rates of SGD depend on the regularity of the optimum and the feature vector.

Weaknesses: N.A.At first sight the setting appear deceptively simple. In practice it applies to a very wide range of application scenarios in which perception is involved

Correctness: Yes

Clarity: The paper is written very well.

Relation to Prior Work: Yes

Reproducibility: Yes

Additional Feedback:

[Author Response · NeurIPS 2020]

Dear Reviewers, we would like to take this opportunity to thank you for your precise and constructive feedback. Below
the paragraphs in italics are extracts from the reviews. Citations refer to the bibliography of the paper.

**Reviewer #1.** *[...] the paper mentions "some basic vision or sound recognition tasks" (line 33) - I'd like to ask some*
*examples of such tasks.*

**Authors:** Our noiseless assumption approximately holds for supervised learning tasks with little ambiguity of the output
given the input - but the rule giving the output given the input can be complex. An example from [18, Section 6] is the
classification of images of cats versus dogs. For typical images, the output is unambiguous; humans indeed achieve a
near-zero error. In sound recognition, one could think of the recovery of the melody from a tune, an unambiguous (but
tremendously complex!) task. We will add these examples to the final version. Note that in Appendix D, we generalize
our results to the case where some ambiguity (i.e., some additive noise) is present.

**R1:** *Looking at the statement of the Theorem 1, seems that it should be applicable in finite-dimensional spaces with*
*invertible covariance matrices. [...] In particular, for X distributed with a finite support and has identity covariance*
*matrix, the conditions (a) and (b) hold for arbitrarily large positive $\alpha$, however the theorem statement implies that the*
*estimates will go to zero at an arbitrarily large polynomial rate, which is not true.*

**A.: Theorem 1 does apply in finite-dimensional spaces.** In the example described by the reviewer, SGD converges
exponentially; this is a surprising effect of the noiseless model. Indeed,
$\mathbb{E}[\|\theta_n - \theta_*\|^2] = \mathbb{E}[\|(I - \gamma X_n \otimes X_n)(\theta_{n-1} - \theta_*)\|^2]$
$\leq \mathbb{E}[\|\theta_{n-1} - \theta_*\|^2] - 2\gamma \mathbb{E}[\langle \theta_{n-1} - \theta_*, (X_n \otimes X_n)(\theta_{n-1} - \theta_*)\rangle] + \gamma^2 R_0 \mathbb{E}[\langle \theta_{n-1} - \theta_*, (X_n \otimes X_n)(\theta_{n-1} - \theta_*)\rangle]$.
As $\gamma R_0 \leq 1$ and by assumption of the reviewer, $\mathbb{E}[X_n \otimes X_n] = I$, we obtain $\mathbb{E}[\|\theta_n - \theta_*\|^2] \leq (1-\gamma)\mathbb{E}[\|\theta_{n-1} - \theta_*\|^2]$.
Note that in finite dimensional spaces, the non-asymptotic polynomial bounds of Theorem 1 can be better than the
exponential rates, for small number of iterations $n$. This is detailed in the paper for the gossip process (lines 288-298).
We will add this remark on the application to finite-dimensional spaces to the final version.

**R1:** *The paper does not give any theoretical argument to the 'tightness' of the proposed bounds, [...].*

**A.:** We prove both upper bounds (Theorems 1 & 3) and lower bounds (Theorems 2 & 4) on the performance of SGD
that almost match: they have the same asymptotic in $n$. Thus the bounds describe the actual behavior of SGD, and this
is confirmed by simulations: **the bounds are "tight" in this sense**. Note that we do not mean "optimal" here.

**R1:** *I would suggest to add the results with additive noise assumption for infinite-dimensional spaces, to put the*
*proposed model into a perspective.*

**A.:** We will give the non-parametric optimal rates with additive noise from Caponnetto & De Vito [9], reached by ridge
regression. The perspective with our work is that these rates are slower than $n^{-1}$, while we prove rates faster than $n^{-1}$
because of our noiseless assumption. This is stated lines 65-68.

**A.:** We also thank **R1** for pointing out a typo and a notation without definition.

**Reviewer #2.** *I think the claim about the Sobolev smoothness is overstatement because it depends on a somewhat*
*strong condition (line 215-216) which strongly restricts the class of kernels.*

**A.: We respectfully disagree.** It is true that this condition does not cover $C^\infty$ kernels, including the Gaussian kernel.
However, **this condition is relevant for less regular kernels**, that have a power decay in Fourier. Line 215-216 defines
the rate of decay in Fourier. In theory, one could imagine that the lower and upper bounds hold for different $s$, in which
case one could have a theory by adapting lines 215-232. However, the point of the section is only to illustrate our theory,
and for all "less regular" kernels that we know and cite, the condition 215-216 holds, so we kept things simple. We will
add this discussion to the final version.

**R2:** *How does the averaging technique work in the noiseless setting?*

**A.:** Averaging does not seem to accelerate the averaging process (Section 3.2). Extrapolating to all SGDs, we expect
that averaging is useful only for reducing additive noise, and thus would not accelerate in the noiseless setting. However,
this question deserves a rigorous study that we wish to conduct in future work.

**Reviewer #2 and Reviewer #4** both asked for a deeper comparison with [18]. [18] indeed analyses the zero (or low)
noise setting, and allow for the optimal function to lie in the kernel space. However, they do not exploit when the
function is more regular than being in the kernel space, i.e., when $\alpha_1 > 0$ with our notation, $\beta > 1/2$ with theirs. In
fact, they leave this case as an open problem in their Section 6. Thus, a fair comparison can only be made when $\alpha_1 = 0$,
$\beta = 1/2$. In this case, SGD and [18] both achieve the same rate $O(n^{-1})$. We will add this discussion to lines 71-73.

**Reviewer #4.** *Line 35: Citation for when it is called multiplicative noise would be great.* **A.:** We will cite [13]. Thanks.

**Reviewer #5** did not express any concern; we only thank her/him for the encouraging review.

[Meta-Review · NeurIPS 2020]

The submission considers noiseless (/low noise) linear regression with non-linear transformation of the input data, and show that under this setting, SGD achieves faster convergence rates. This is a very nice contribution with applicability to important problems as mentioned by the authors in their feedback. We urge the authors to incorporate the points they made in response to the reviews.